# Nondestructive X-ray tomography of brain tissue ultrastructure

Carles Bosch [1], Tomas Aidukas [2], Mirko Holler [2], Alexandra Pacureanu [3], Elisabeth Müller [4], Christopher J. Peddie [5], Yuxin Zhang [1,6], Phil Cook [3], Lucy Collinson [5], Oliver Bunk [2], Andreas Menzel [2], Manuel Guizar-Sicairos [2,7], Gabriel Aeppli [2,7,8,9], Ana Diaz [2]✉, Adrian A. Wanner [4]✉ & Andreas T. Schaefer [1,6]✉

Maps of biological tissues at subcellular detail are key for understanding how organs function. X-ray nanotomography is a promising alternative to volume electron microscopy: it has the potential to nondestructively image millimeter-sized samples at ultrastructural resolution within a few days. A fundamental barrier is that the intense X-rays required for imaging also deform and disintegrate the tissue samples. Here we show a combination of solutions that overcome this barrier: We used a cryogenic and stable sample stage, tailored nonrigid tomographic reconstruction algorithms and an epoxy resin developed for the nuclear and aerospace industry. Tissue samples were resistant to radiation doses exceeding $1.15 \times 10^{10}$ Gy, and sub-40 nm isotropic resolution allowed identifying axon bundles, dendrites and synapses in mouse brain tissue without physical sectioning. Using volume electron microscopy, we demonstrate that tissue ultrastructure remains intact after X-ray imaging. Together, this unlocks the potential of X-ray tomography for high-resolution tissue imaging.

Current gold-standard methods for dense tissue reconstruction rely on volume electron microscopy[1–3]. As electron penetration depth into tissue is limited to, at most, a few hundred nanometers[4,5], obtaining three-dimensional (3D) information requires tissue sectioning or milling either before preparing samples for imaging[6] or during image acquisition[7,8]. Such workflows present challenges for reliable continuous image acquisition, with the risk of information loss during month-long experiments, and an experimental analysis that requires reconstruction and alignment of multimodal datasets.

X-rays can penetrate samples of several millimeters to centimeters, allowing for largely nondestructive imaging of tissues[4,5]. Laboratory X-ray sources can resolve cell bodies and even large neurites[9] but their limited brilliance makes high-resolution imaging extremely

time consuming. Large-scale synchrotron facilities can provide high-brilliance X-rays at wavelengths in the Ångstrom range, and are therefore driving the development of imaging methods capable of providing 3D resolution down to nanometers for hard, inorganic nanostructures with high-density contrast[10–16]. The application of these methods to biological imaging has enabled the reliable identification of various biological features including blood vessels, cell bodies and large neurites[17–20]. Notably, thin neurites were identified at a resolution of ~100 nm on a sample volume of about $100 \times 100 \times 100$ μm³ (ref. 21) using nano-holotomography[22] with an instrument dedicated to high-resolution X-ray imaging in cryogenic conditions. Fundamentally, however, the resolution in X-ray tomography scales with the fourth power of the radiation dose; that is, to increase the resolution

¹Sensory Circuits and Neurotechnology Lab, The Francis Crick Institute, London, UK. ²Paul Scherrer Institute PSI Center for Photon Science, Villigen, Switzerland. ³European Synchrotron Radiation Facility, The European Synchrotron, Grenoble, France. ⁴Paul Scherrer Institute PSI Center for Life Sciences, Villigen, Switzerland. ⁵Electron Microscopy STP, The Francis Crick Institute, London, UK. ⁶Department of Neuroscience, Physiology and Pharmacology, University College, London, UK. ⁷Institute of Physics, École Polytechnique Fédérale de Lausanne, Lausanne, Switzerland. ⁸Department of Physics, Eidgenössische Technische Hochschule Zurich, Zurich, Switzerland. ⁹Quantum Center, Eidgenössische Technische Hochschule Zurich, Zurich, Switzerland. ✉e-mail: ana.diaz@psi.ch; adrian.wanner@psi.ch; andreas.schaefer@crick.ac.uk

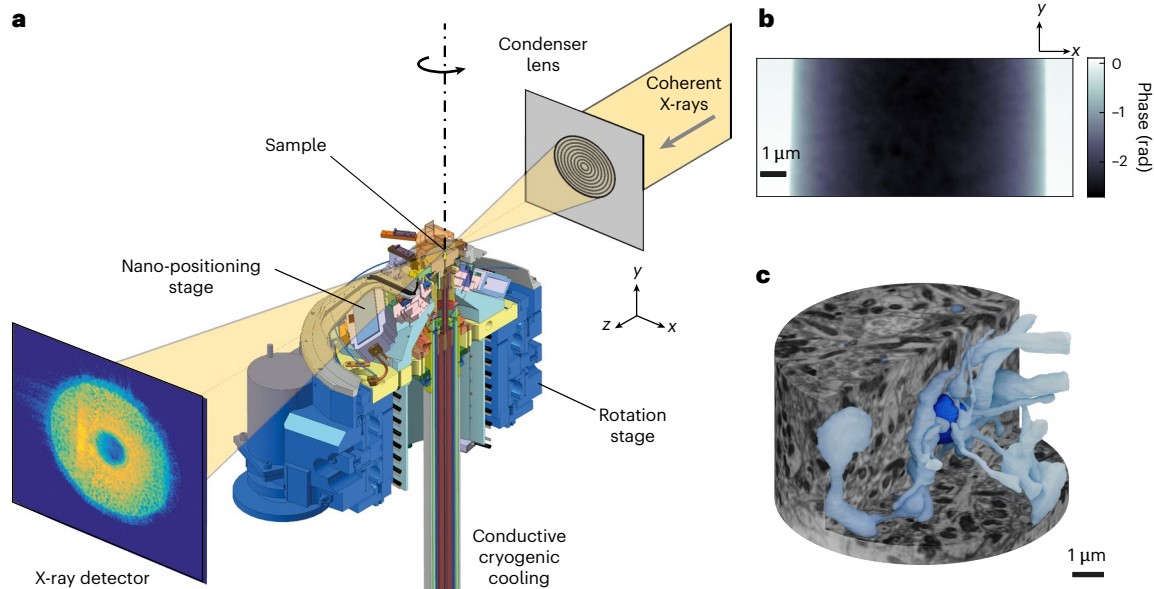

**Fig. 1 | X-ray ptychography of brain tissue. a**, The ptychography setup: the incoming coherent beam is focused by a lens to define the illumination on the sample (top right), a cryogenic stage for scanning with 10 nm accuracy with rotation capability (center) and an example diffraction pattern recorded by the X-ray detector (bottom left). **b**, A single reconstructed phase projection of a resin E (TGPAP)-embedded mouse brain tissue sample extracted from the external plexiform layer in the olfactory bulb, obtained using a monochromatic 6.2 keV beam under cryogenic conditions at cSAXS. **c**, A 3D rendering of the reconstructed tomogram containing the projection shown in **b** (1,190 projections, total absorbed dose $2.5 \times 10^9$ Gy). A part of the volume has been cut to show the segmentation of a neurite (dark blue) and some of its presynaptic partners (light blue), based on the X-ray tomogram dataset.

by a factor of 5 in the same sample, the X-ray dose needs to be increased by a factor of 625. X-ray imaging of soft biological tissue was previously thought to be limited by tissue contrast and by the radiation dose that can be employed before the tissue is damaged[4,5,23], and it remained unclear whether key ultrastructural features could be examined with X-rays at sufficient resolution.

To overcome the limited X-ray image contrast inherent to soft tissues, we combine heavy metal staining protocols developed for volume electron microscopy[24] with high-resolution ptychographic tomography[10,25]. Radiation damage is minimized by imaging samples under cryogenic conditions using a cryogenic ptychographic tomography instrument[26,27] featuring sub-25 nm accuracy scanning with a rotational degree of freedom achieved by laser interferometry[28]. The effects of the radiation on the samples are further minimized by embedding them in a highly radiation-resistant resin[29,30]. Residual sample deformations will introduce inconsistencies between the acquired projections at length scales below 60 nm, for which we compensate computationally to avoid loss of 3D reconstruction quality[31]. By combining these techniques, we obtain an isotropic 3D resolution of 38 nm for neural tissue, sufficient for resolving synapses while simultaneously maintaining tissue integrity.

## Results

### X-ray ptychography resolves brain tissue ultrastructure

X-ray ptychography is performed by scanning a sample across a confined, coherent beam a few micrometers in diameter and measuring diffraction patterns in the far field (Fig. 1a)[32]. Using iterative reconstruction algorithms[33,34], these diffraction patterns (Fig. 1a) are then used to produce phase images, which are proportional to the projected electron density of the sample (Fig. 1b). To perform ptychographic X-ray computed tomography (PXCT), the projections are acquired at multiple sample rotation angles from 0° to 180°, followed by tomographic reconstruction of a 3D sample volume (Fig. 1c). Resolution in X-ray ptychography is limited not by the beam size but by the angular acceptance of the detector, which determines the maximum observable spatial frequencies, and the applied X-ray dose, which determines the

signal-to-noise ratio of the spatial frequencies within the recorded diffraction patterns. Since the imaging geometry can be adjusted relatively easily, radiation damage and weak intrinsic scattering of biological tissues are the main factors limiting the attainable resolution[5,23,35]. We therefore employed a cryo-tomography stage[26] to cool samples to <95 K and to reduce the structural changes caused by ionizing radiation, in combination with a staining and embedding protocol to maximize contrast. In addition, we applied a nonrigid tomography reconstruction algorithm designed to compensate for radiation-induced deformation of the sample[31]. In the following, we present tomograms from mouse brain tissue (the radiatum layer of hippocampus, the external plexiform and glomerular layers from the olfactory bulb, and upper layers of neocortex as structures rich in neurites and dendritic shaft and spine synapses) at different radiation doses.

The resolution of PXCT (and more generally X-ray tomography) scales with the fourth power of the applied dose[23]. Increased dose, in turn, can result in sample deformation and ultimately the destruction of the sample. To assess these radiation limits we stained mouse olfactory bulb tissue with a (ferrocyanide) reduced osmium–thiocarbohydrazide–osmium (ROTO) stain[24] and embedded it in a standard 'hard EPON' resin[36] (hard EPON resins A and B; Fig. 2a, Extended Data Fig. 1 and Supplementary Tables 2 and 3). We then exposed samples to a broad bandwidth beam with an average energy of 25 keV, generated by a bending magnet at the European Synchrotron Radiation Facility (ESRF) beamline BM05 (ref. 37), leveraging the high flux of the 'extremely bright source', the recently upgraded ESRF fourth-generation storage ring. Indeed, samples began to readily show mass loss and disintegrated fully with sustained exposure (Fig. 2b and Supplementary Video 1).

To improve radiation resistance, we turned to resins previously developed for nuclear reactors or the aerospace industry[29,30,38], optimized for stability in the presence of ionizing radiation. Screening a battery of such resins, we identified a low-viscosity epoxy with good infiltration properties consisting of triglycidyl *para*-aminophenol (TGPAP) (Fig. 2c), a triglycidyl ether of para-aminophenol. Like EPON812, TGPAP monomers are trifunctional; that, is they contain three epoxy groups, but also a benzene ring and a tertiary amine group.

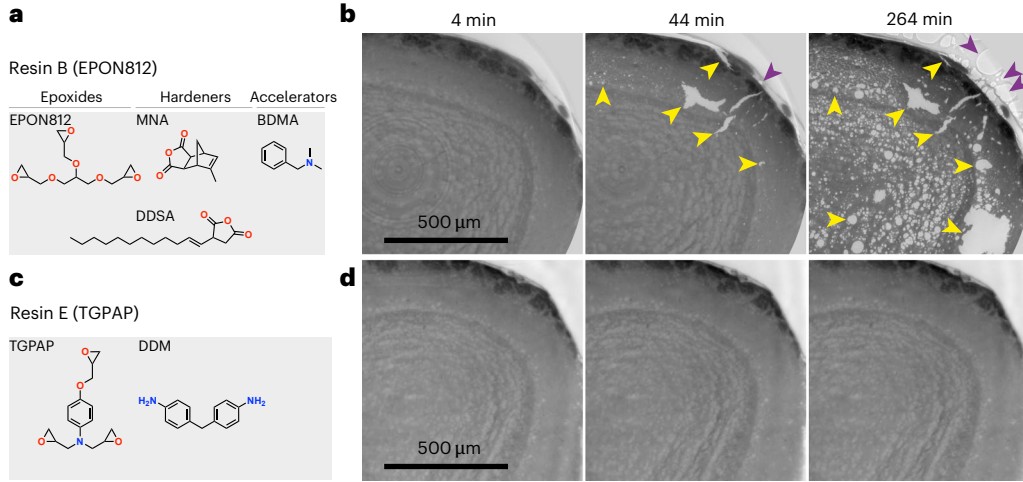

**Fig. 2 | A 'tough' epoxy resin for increased radiation resistance. a**, The chemical structure of the monomers used to polymerize resin B (EPON812-MNA-DDSA-BDMA, later referenced as 'EPON812' for simplicity), one of the most representative resins explored in this study. **b**, Radiation exposure tests on stained olfactory bulb samples embedded in resin B (EPON812), performed at room temperature and ambient pressure using a filtered polychromatic X-ray beam with average energy of 25 keV (ESRF BM05). The assessment of radiation effects is based on phase-contrast microtomographic imaging. Note the resin degradation within (yellow arrowheads) and outside (purple arrowheads) the embedded tissue. **c**, As in **a**, but for the radiation 'tough' resin E (TGPAP-DDM, later referenced as 'TGPAP' for simplicity). **d**, The same as **b** for samples embedded in resin E (TGPAP).

For curing, we used 4,4′-diaminodiphenyl methane (DDM), a well-known hardener with two primary amine groups that react with the epoxy groups of TGPAP (Extended Data Fig. 1 and Supplementary Tables 2 and 3). The three epoxy groups per molecule allow for increased crosslinking during curing, leading to superior mechanical strength and chemical and radiation resistance, as well as thermal stability. Moreover, the benzene rings in both the monomer and the hardener are thought to further increase radiation resilience. When exposing samples embedded in TGPAP to the ESRF beamline BM05 (ref. 37) they showed little sign of mass loss or compromise of cellular features even with prolonged exposure (Fig. 2d and Supplementary Video 1). Embedding heavy metal-stained tissue in TGPAP should therefore allow an increase of the cumulative radiation dose for PXCT.

To assess what brain structures PXCT can resolve, we prepared pillars from mouse olfactory bulb tissue stained with ROTO and embedded in either hard EPON resins or TGPAP–DDM, to be measured at the coherent small-angle X-ray scattering (cSAXS) beamline of the Swiss Light Source (SLS). To allow for time-efficient imaging with the third generation SLS, that is, before the upgrade to a fourth-generation synchrotron ring, we focused ion beam-milled pillars with a diameter of 10–30 μm. At a dose of $1.8 \times 10^7$ Gy we were able to readily resolve dendrites in a sample with a volume of about $20 \times 20 \times 20$ μm² (Fig. 3a and Extended Data Fig. 2a); using Fourier shell correlation (FSC; Extended Data Fig. 2c)[39] we estimated an isotropic 3D resolution of 84 nm (Extended Data Fig. 2c,d). Cryo-cooling is thought to reduce structural changes that limit resolution in X-ray imaging due to radiation damage[5,40]. Consistent with this notion, acquired tomograms did not show any indication of blurring or other signs of structural changes due to radiation damage. Irradiating a sample with an increased radiation dose of $8.4 \times 10^7$ Gy resulted in a 3D resolution of 69 nm (Extended Data Fig. 2a,c,d). However, contrary to theoretical predictions[35], an even higher dose of $3.8 \times 10^8$ Gy did not enable an enhanced resolution (70 nm; Extended Data Fig. 2a,c,d). In particular, the sample deformed, presumably due to 'softening' of the embedding resin, as seen when comparing low-resolution tomograms acquired with ~$2.6 \times 10^8$ Gy intermittent irradiation (Extended Data Fig. 3). As a result, the resolution of the combined tomogram decreased based on visual blurring in the 3D volume and the decrease of the correlation at several spatial frequencies, as determined by FSC analysis (Extended Data Fig. 2a,c). To take these sample changes into account,

we applied a nonrigid tomography reconstruction approach that estimates the deformations between the subtomograms and includes them in the tomographic reconstruction[31] (Extended Data Fig. 3). This deformation correction method requires tailored data acquisition where the whole angular range from 0° to 180° is split into multiple equally spaced angular ranges to acquire multiple subtomograms. In doing so, each subtomogram provides a low-resolution snapshot of the sample at a given time point. By calculating the 3D optical flow between these subtomograms, the deformation fields can be estimated across a range of time points and interpolated to describe the changes between all collected projections. Accounting for radiation-induced changes in this manner recovered higher spatial frequencies (Extended Data Figs. 3 and 2b,c), which translated into an improved 3D resolution of 49.7 nm (Fig. 3b,d and Extended Data Fig. 2b–d), that is, close to the theoretical predictions of resolution improving with the fourth power of dose[23]. The use of nonrigid tomography consistently improved the expected scaling between the radiation dose and attainable resolution across the measured samples (Fig. 3d and Extended Data Fig. 2d). Measured at a temperature of ~90 K, tissue was radiation stable at doses up to $6 \times 10^8$ Gy (Extended Data Fig. 4) when embedded with either of the traditional electron microscopy (EM)-embedding resins EMbed812 or EPON812[41] (Extended Data Figs. 1 and 5).

Further increasing the dose beyond $6 \times 10^8$ Gy with hard EPON samples consistently resulted in radiation damage such as widespread mass loss with the monochromatic 6.2 keV beam (Extended Data Fig. 5b–e) as previously observed with broadband 25 keV irradiation at beamline BM05 (Fig. 2b). Leveraging the radiation-resistant resin TGPAP–DDM, however, allowed us to increase the applied dose further. Using TGPAP for embedding (with the same ROTO staining and mouse olfactory bulb tissue) we found no indication of mass loss and only localized submicron punctae that seemed to emerge with a total dose of >$10^9$ Gy but did not impact on relevant tissue features (Extended Data Fig. 5g–j). This increased dose enabled us to further increase the resolution to 38 nm on a sample with a volume of about $10 \times 10 \times 5$ μm³ (Fig. 3c–e and Supplementary Video 2).

Using this improved approach with imaging ROTO-stained samples at cryogenic temperatures with nonrigid tomographic reconstructions, we obtained 3D tomograms for ROTO-stained tissue from a variety of brain regions, including the olfactory bulb external plexiform layer, glomerular layer, hippocampus stratum radiatum and layer two

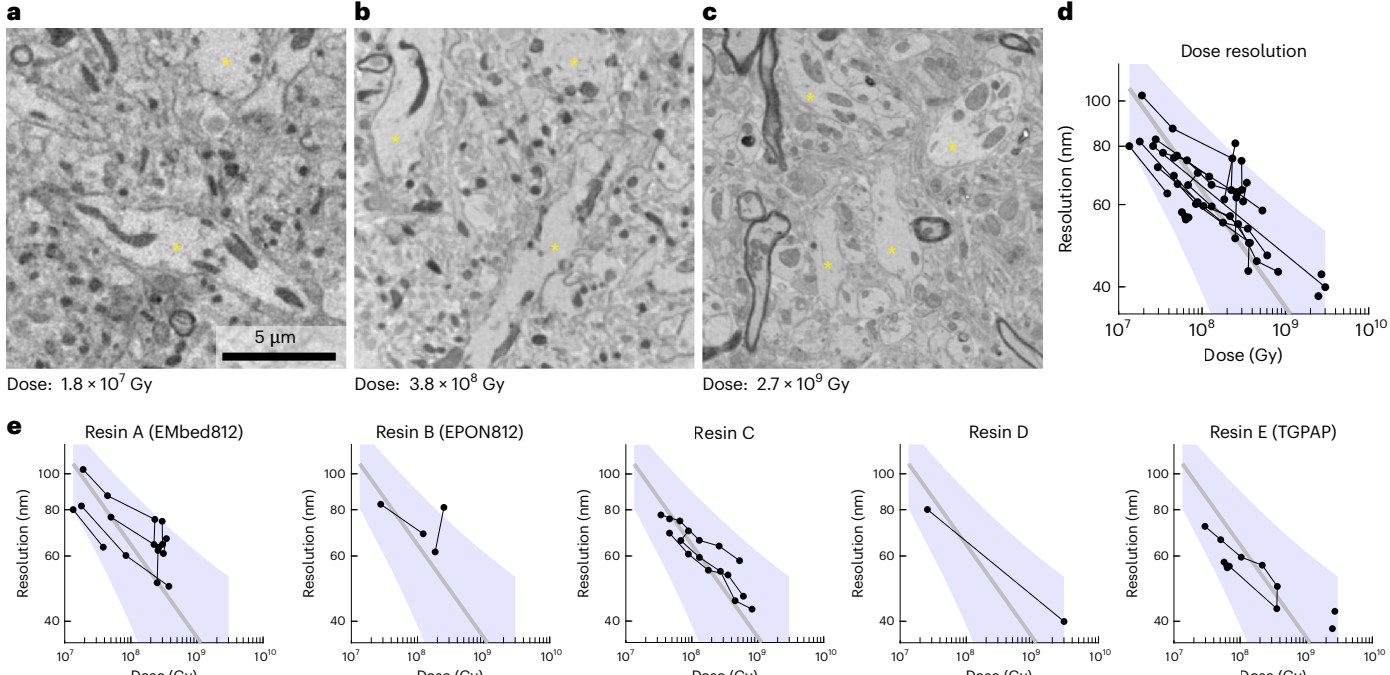

**Fig. 3 | Dose dependence of resolution for X-ray ptychography of brain tissue.**
**a**–**c**, Example 2D slices from 3D nonrigid reconstructions of tomograms acquired at increasing radiation doses ($1.8 \times 10^7$ Gy (**a**), $3.8 \times 10^8$ Gy (**b**) and $2.7 \times 10^9$ Gy (**c**)) of samples embedded in resin A (EMbed812) (**a** and **b**) and resin E (TGPAP) (**c**). The yellow asterisks indicate dendrites of projection neurons in all three images. **d**,**e**, FSC resolution for all tomogram series of all samples reported, imaged at different doses and not displaying widespread mass loss. Panel **e** shows all tomograms in **d**, split into different plots depending on the resin samples were embedded in. The tomograms (circles) obtained from the same sample at multiple doses form a series and are linked with a black edge. A power function FSC_resolution = $a$/dose$^{(1/4)}$ fitted to all plotted data points is shown (gray line) along with its 95% prediction intervals with observational nonsimultaneous bounds (shaded purple area). The shaded area is calculated for the full set of tomograms in **d** and displayed as reference in **e**.

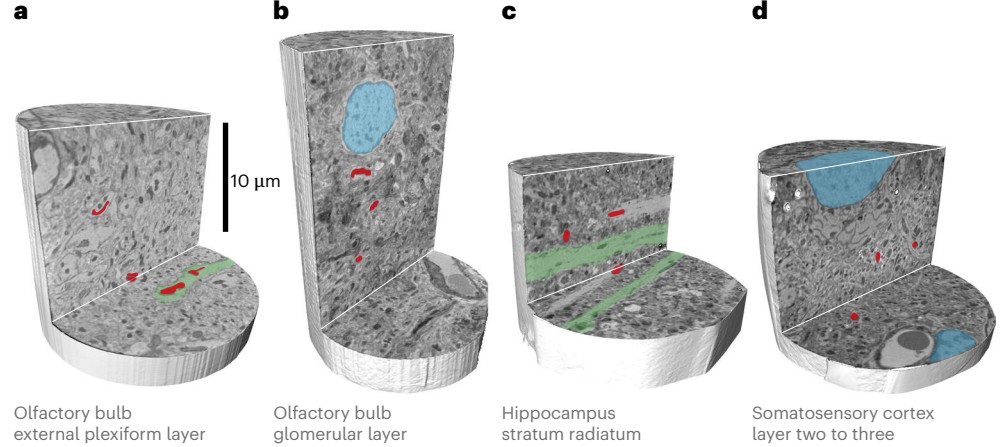

**Fig. 4 | Subcellular features detectable with X-ray ptychography in different brain tissues. a–d,** Cut-out views from X-ray ptychography tomographic reconstructions for different tissues (imaged using radiation doses of $3.8 \times 10^8$ Gy, $3.8 \times 10^8$ Gy, $2.2 \times 10^8$ Gy and $1.9 \times 10^8$ Gy): the respective tissue samples (olfactory bulb external plexiform layer (**a**), olfactory bulb glomerular layer (**b**), hippocampus stratum radiatum (**c**) and layer two to three of the somatosensory cortex (**d**)) were embedded in resin A (EMbed812) (**a**–**c**) and resin B (EPON812) (**d**). The subcellular features including dendrites (green), nuclei (blue) and mitochondria (red) are resolved.

to three of the somatosensory cortex (Fig. 4). This allowed us to visualize, for example, neurites, mitochondria, myelin or nuclei (Fig. 4).

### Synapse detection with X-rays
A critical feature for connectomics is the detection of synaptic contacts (Fig. 5). Thus, we assessed whether PXCT would allow the identification of such structures. Indeed, for a dose of $3.8 \times 10^8$ Gy in EPON-embedded olfactory bulb samples, structures resembling synapses became apparent (Fig. 5a,e). PXCT being a largely nondestructive technique, we

subsequently processed the same specimen using focused ion beam milling combined with scanning electron microscopy (FIB–SEM). This allowed us to obtain insights into potential ultrastructural alterations induced by the X-ray irradiation as well as obtain a 'ground truth' for structure identification and further quantification. The FIB–SEM reconstruction showed a mild curtaining effect that might be the consequence of radiation-induced changes to the embedding medium (Extended Data Fig. 6). However, the overall ultrastructure was not noticeably impacted by prior X-ray radiation dosage of ~$4 \times 10^8$ Gy

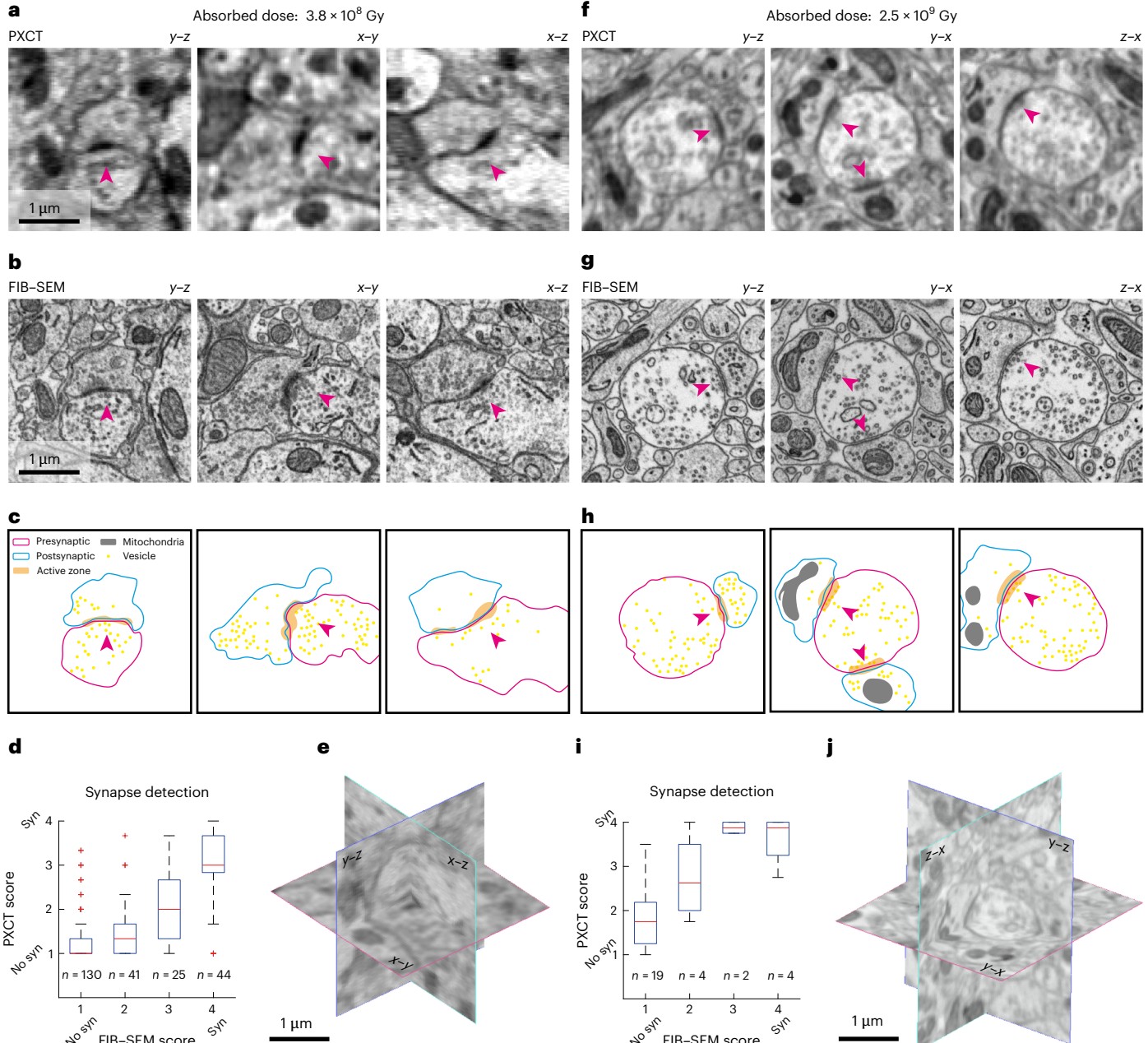

**Fig. 5 | Quantitative evaluation of synapse detectability with correlative X-ray ptychography and FIB–SEM. a**, Example PXCT images of a dendrodendritic synapse (magenta arrowheads) from the external plexiform layer in three orthogonal reslices, acquired from a tissue sample embedded in resin A (EMbed812) imaged with a total absorbed dose of $3.8 \times 10^8$ Gy. The arrowhead points to a synapse. **b**, The same region in **a** from the corresponding FIB–SEM dataset. **c**, Segmented features of the neurites establishing the synapses shown in **b**. **d**, The presence or absence of synapses (syn) was evaluated in $n = 240$ randomly selected $1\,\mu m^3$ subvolumes in both FIB–SEM and PXCT volumes.

The box plot indicates the median score given to the distinct regions of interest, binned by their EM scores. The box covers the 25–75% percentile range, the middle bar represents the median value and the whiskers extend to the most extreme data points that are not an outlier (defined as outside of the 1.5× interquartile range). **e**, Spatial distribution of the orthogonal planes shown in **a**. **f–j**, Same as in **a–e**, for a tissue sample embedded in resin E (TGPAP), imaged with a total absorbed dose of $2.5 \times 10^9$ Gy. In this case the analysis was performed based on $n = 29$ subvolumes.

(Fig. 5b). In particular, membranes were left intact, as were subcellular features such as mitochondria, endoplasmic reticula or vesicles (Fig. 5b and Extended Data Fig. 6). Aligning the FIB–SEM with the PXCT dataset allowed us to directly compare structures identified in PXCT with the ground truth data obtained from FIB–SEM (Fig. 5a–c). To quantify the reliability of synapse identification, we performed two approaches: first, we traced individual dendrites and performed independent manual labeling of synapses in both datasets (Extended Data Fig. 7a). More than two-thirds of synapses identified in the FIB–SEM dataset

were recognized in a blind analysis of PXCT images (225/338 synapses, 70 ± 6%, $n = 5$ dendrites; Extended Data Fig. 7a,b).

To obtain true estimates of precision and reliability, annotators were presented with randomly sampled $1\,\mu m^3$ volumes of the FIB–SEM or PXCT datasets and asked to indicate whether a synapse was present in the volume with a confidence score (Extended Data Fig. 8). Using the FIB–SEM dataset as ground truth again, PXCT annotation showed a recall of synapse presence of 65% with a precision of 80% (Fig. 5d and Extended Data Fig. 8g,h).

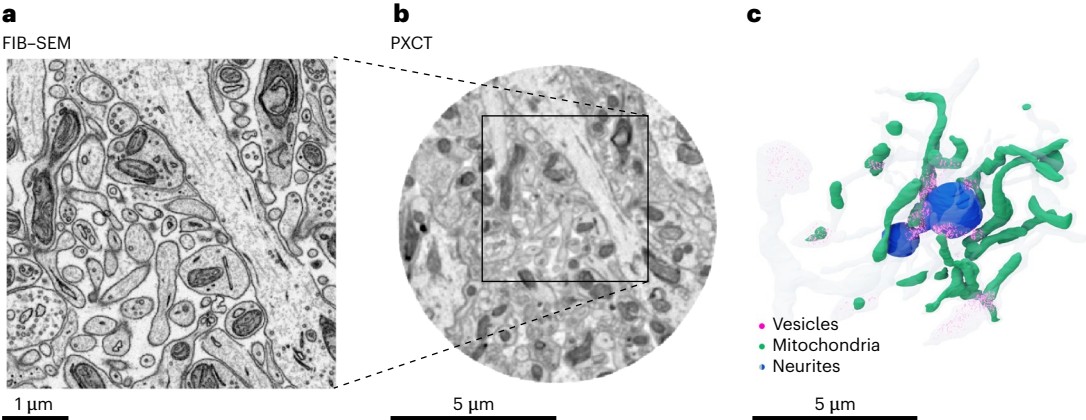

**a** FIB–SEM

**b** PXCT

**c**

1 µm

5 µm

5 µm

• Vesicles
• Mitochondria
• Neurites

**Fig. 6 | Tissue ultrastructure can withstand X-ray doses of 2.5 × 10⁹ Gy. a,b**, The infiltration quality and the ultrastructural integrity for the TGPAP-embedded samples was assessed at high resolution using FIB–SEM (**a**) after PXCT (**b**). **c**, Some neurites segmented manually using the PXCT data only. A dark blue neurite receiving synaptic contacts from transparent pale blue neurites is shown alongside mitochondria (green) and vesicle clusters (magenta).

We then applied the same approach to the higher dose, TGPAP–DDM-embedded samples, performing FIB–SEM milling on the (smaller 10 µm) pillar after PXCT measurements. While the smaller volume contained fewer synaptic features (Extended Data Fig. 8d), identification of synapses in the PXCT dataset indeed tended to be further improved (Fig. 5i and Extended Data Fig. 8i–k).

Importantly, performing FIB–SEM analysis after PXCT allowed us to further assess the properties of the TGPAP–DDM resin. Notably, it showed that TGPAP tissue penetration is comparable to commonly used EPON and the ultrastructure is well preserved (even after irradiation with a total dose exceeding $1.15 \times 10^{10}$ Gy; Fig. 6a,b). The extracellular space comprises 16 ± 2% (mean ± s.e.m.) (Supplementary Fig. 1), comparable to other sample preparation protocols[42,43]. It also allowed us to attempt segmentation of different ultrastructural features. Resolution was sufficient to resolve and manually segment vesicle clusters and synaptic densities (Fig. 6c). Automatic segmentation with segmentation pipelines developed for EM data showed that, for example, high-contrast features such as mitochondria as well as many segmented neurites were consistent with the corresponding segmentation of the FIB–SEM dataset (Extended Data Fig. 9b,c). Thinner neurites showed some splits and mergers (Extended Data Fig. 9c–f), indicating that further resolution improvement and possibly segmentation tailored to the PXCT dataset (possibly leveraging its nondestructiveness) will be beneficial for reliable automatic segmentation of all relevant connectomics features[44].

Together, this indicates that using radiation-resistant embedding materials with heavy metal staining and nonrigid tomographic reconstruction methods, X-ray nanotomography can provide sufficient 3D resolution to identify many key connectomic features.

## Discussion

Here, we have demonstrated the potential for synchrotron X-ray nanotomography to image at resolutions sufficient to reliably detect key ultrastructural features such as synaptic contacts. While employing sub-nm wavelengths, resolution in X-ray imaging of biological soft tissue is generally limited by contrast. Furthermore, the softness of embedding materials raises susceptibility to radiation damage, which ultimately limits the maximum attainable resolution. When heavy metal stains are employed, they increase contrast and X-ray absorption. Absorption, however, underlies radiation damage[4,5,23]. At the SLS, we employed the tOMography Nano crYo stage (OMNY), an instrument that uses an interferometer-guided cryogenic stage optimized for PXCT measurements[26]. The OMNY stage[26] allows for cooling to both liquid nitrogen and liquid helium temperatures. Cooling to the latter

lower temperatures might further reduce radiation damage, allowing for additional increases in radiation dose and resolution.

Embedding and staining protocols have played a key role in the development and improvement of volume EM techniques[24,45–47]. EPON embedding provides excellent mechanical properties for thin serial sectioning[48] but is prone to radiation damage[49]. By contrast, TGPAP–DDM penetrated tissue well, preserving ultrastructure and showed improved radiation resistance. Therefore, we call the TGPAP polymer 'tough resin'. Theoretical estimates for flux and dose constraints for X-ray tomographic imaging indicate that such dose resistance will suffice for whole-mouse brain X-ray tomographic imaging at sub-20 nm resolution[35]. TGPAP in combination with PXCT might be applicable for other sample preparation approaches without initial chemical fixation as well. For example, high-pressure frozen[50] or frozen-hydrated[27] samples have been imaged with PXCT and such samples could be embedded in TGPAP before imaging to improve radiation resistance and enable higher-resolution imaging.

We have used PXCT to obtain nanometer-resolution volumetric data without sectioning. Nevertheless, it is useful to perform conventional volume electron microscopy on TGPAP-embedded samples; for example, to obtain high-resolution 'ground truth' data for subvolumes. Ultrathin sectioning might need further optimizing of the resin and thorough quantification[48]. Since we optimized the choice of resin for high-dose X-ray imaging, it is likely that for pure FIB–SEM or serial sectioning experiments other resins might be the better choice.

While not resulting in tissue disintegration or ultrastructural damage, irradiation of samples may still result in localized tissue alterations as well as global deformations, probably due to thermal expansion or limited impact on chemical bonds and, for example, local gas formation. The ultrastructure (for example, membranes and protein clusters) was largely unimpacted; however, such deformations result in blurred tomographic reconstructions, limiting resolution. We therefore employed nonrigid reconstruction methods that partially mitigate these limitations. Improvements of such algorithms or tomographic acquisition with intermediate, low-dose reference tomograms, might further improve resolution to levels below 20 nm. We note that, for connectomics, deformations of the sample caused by radiation-induced changes or by sample preparation are not limiting as long as the neural connections can be reliably tracked. Correlative volume EM datasets can provide ground truth for the development of such algorithms. While providing good contrast, heavy metal stains might, in fact, not be ideal for X-ray imaging, and materials with similar scattering but reduced absorption[51] might further improve sample stability.

Correlative volume EM can also aid downstream data processing and segmentation by providing a higher-resolution ground truth for training data. Moreover, with algorithms emerging to predict for example the chemical type of a synapse from volume EM data[52], increased resolution as well as ground truth data will help the further annotation of X-ray data.

There is no fundamental limit to scale up PXCT to larger volumes without the need of physical stitching[35], yet it presents several challenges. First, in large volumes and at high resolution, X-ray penetration depth is larger than the depth of field. With the increased longitudinal coherence enabled by fourth-generation synchrotron sources, 'multislice' ptychographic reconstruction methods can, however, explicitly account for such beam propagation effects[53–56]. This not only provides an efficient path to large volume X-ray imaging, but also relaxes the Crowther criterion, reducing the required number of rotation angles[56]. Second, one practical challenge of using PXCT for neural circuit imaging could be the high photon energy needed to penetrate metal-stained samples of several millimeters, because at these energies it is more difficult to achieve high brilliance at synchrotron sources and detectors are not as efficient. While more tailored staining protocols might circumvent these limitations, changing the imaging geometry is an attractive alternative to increase the accessible volumes with current technology. In a laminography geometry[11,57,58], the axis of rotation is tilted compared with tomography. This allows imaging of millimeter (or, in principle, centimeter) wide and ~10–50-μm-thick slabs with <20 nm resolution for nonbiological substrates (using ptychography as well[11]). For embedded biological tissue, such slices can be produced using hot-knife techniques[59], currently employed in preparation for FIB–SEM imaging. The versatility of synchrotron X-ray imaging then allows to rapidly acquire overview images followed by repeated targeted high-resolution image acquisition[11].

Herein, we show that X-ray coherent imaging can densely resolve key connectomic features such as synapses in tissue. Importantly, we demonstrate that biological tissue can be prepared and imaged to withstand radiation dose exceeding $1.15 \times 10^{10}$ Gy, removing the fundamental limitations to employing X-ray tomography for ultrastructural-resolution tissue imaging. The advent of fourth-generation, high-brilliance X-ray sources will introduce this fundamentally nondestructive, scalable and highly reliable technique as a widely available, powerful component of the connectomics and wider tissue life sciences toolbox.

## Online content

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

## Methods

All samples, tomogram series, doses and resolutions of all individual tomograms and hyperlinks to all 3D datasets are listed in Supplementary Table 1. A summary of all samples is listed in Supplementary Table 5.

### Animals

Animals used in this study were 6–16-week-old wild-type mice of C57Bl/6 or CD-1 background of either sex. Mice were housed up to five per cage with an ad libitum supply of food and water under a 12–12 h light–dark cycle at an ambient temperature of $22 \pm 2\,°C$ and a relative humidity of $55 \pm 10\%$. All animal protocols applying to all mice except M1 were approved by the Animal Welfare and Ethical Review Body of the Francis Crick Institute and the UK Home Office under the Animals (Scientific Procedures) Act 1986. For M1, animal experiments were approved by the cantonal veterinary office of the canton Aargau (Switzerland) and were carried out in accordance with the Swiss law on animal protection. All animal IDs are listed in Supplementary Table 5.

### Statistics and reproducibility

All images displaying biological features report intuitive examples of features robustly resolved in independent specimens. The resin radiation stress test experiment reported in Fig. 2b,d was run on a single sample per experimental group. The ptychography experiments providing dose and resolution insights on reconstructed tomograms reported in Figs. 1b,c and 3–6 and Extended Data Figs. 1–5 involve a cohesive dataset of reconstructions made available in Supplementary Table 1 and in ref. 60 and the necessary code in ref. 61. This dataset comprises 17 independent physical samples (Supplementary Table 5) differing in diameter, mouse brain region and the resin they were embedded in. Samples were imaged multiple times each, resulting in tomogram series throughout which the accumulated dose could be tracked. A 'tomogram series' was defined when the sample region being imaged was kept constant (within 3 μm precision in sample height, comparable to the diameter of the incident beam) and the irradiated dose was monitored (to at least a $1 \times 10^8$ Gy step). The dataset contains 19 tomogram series, each containing one or more tomograms, adding up to a total of 63 tomograms. All tomograms were reconstructed using a 'rigid' algorithm (63 rigid reconstructions), and all but two tomograms (61) were reconstructed using a 'nonrigid' algorithm (Extended Data Fig. 3), adding up to 124 tomogram reconstructions. The 2D dose-resolution results reported in Extended Data Fig. 10 arise from a dataset comprising five tomogram series (from four samples). Each tomogram series was scanned at different radiation doses by varying exposure time and step size. In two tomogram series (CX1 and EPL8), the acquisition at all doses was replicated seven and six times in each sample, respectively. All data points from all replicates are plotted in Extended Data Fig. 10b.

### Embedding resins

In total, five different embedding polymers were employed. Each resin originates from the mixture of a set of monomers at specific ratios. Then, each resin mixture was polymerized at 70 °C for ~72 h. Their monomer and putative polymerized structure is reported in Extended Data Fig. 1, the monomer identities are listed in Supplementary Table 2 and the ratios are detailed in Supplementary Table 3. An overall nomenclature is used, referring to them as 'A, B, C, D, E'. To enhance readability of the statements, we often simultaneously refer to them with the common name or acronym given to the epoxide monomer (such as 'EPON812'). Finally, resins 'A, B, C' are commonly used in volume EM and often referred to with the overarching term of 'hard EPON', which applies to the specific ratios of hardener:resin used[36]. Accordingly, the term 'hard EPON' has been used in conjunction with the aforementioned terminology to support readability. By contrast, resin 'E' consisting of the epoxy–hardener mixture TGPAP–DDM is referred to as 'tough' highlighting its resilience to radiation.

### Tissue preparation

**Dissection.** Mice were killed and 600-μm-thick sections of brain areas of interest were sliced in ice-cold dissecting buffer (phosphate buffer 65 mM, 0.6 mM $CaCl_2$ and 150 mM sucrose) with an osmolarity of $300 \pm 20$ mOsm $l^{-1}$ using a LeicaVT1200S vibratome and immediately transferred to ice-cold fixative (either 1% glutaraldehyde or 1.25% glutaraldehyde and 2.5% paraformaldehyde in 150 mM sodium cacodylate buffer pH 7.40, $300 \pm 20$ mOsm $l^{-1}$). Samples were left in the same fixative overnight, at 4 °C. The fixative was then washed with wash buffer (150 mM sodium cacodylate pH 7.40, $300 \pm 20$ mOsm $l^{-1}$) three times for 10 min at 4 °C. Overall, samples were kept in an ice-cold, osmolarity-checked buffer. Mouse M1 was euthanized and instantly perfused through the left heart ventricle with ice-cold Ringer solution supplemented with 50,000 U $l^{-1}$ heparin at a flow rate of 12 ml $min^{-1}$. Thereafter, the brain was extracted, cut into <1 $mm^3$ blocks with a blade and fixed in ice-cold fixative (1.3% glutaraldehyde and 2.5% paraformaldehyde in 100 mM sodium cacodylate buffer pH 7.41 with 4 mM $CaCl_2$) overnight.

**Staining, dehydration and embedding.** Slabs were stained with heavy metals using an established ROTO protocol[42] using an automated tissue processor (Leica EMTP). Briefly, they were first stained with reduced osmium (2% $OsO_4$, 3% potassium ferrocyanide and 2 mM CaCl2 in wash buffer) for 2 h at 20 °C, followed with 1% thiocarbohydrazide (aq.) at 60 °C for 50 min, 2% osmium (aq.) for 2 h at 20 °C and 1% uranyl acetate (aq.) overnight at 4 °C. The next day, the samples were further stained with lead aspartate for 2 h at 60 °C. Samples were washed with double-distilled water six times for 10 min at 20 °C between each staining step, except warmer washes before and after thiocarbohydrazide (50 °C) and before lead aspartate (60 °C).

Samples were then dehydrated with increasing ethanol solutions (75%, 90% and 2× 100%), transferred to propylene oxide, and infiltrated with hard EPON mixed with propylene oxide in increasing concentrations (25%, 50%, 75% and 2× 100%). Finally, samples were transferred into plastic molds with freshly prepared resin and polymerized individually for 72 h at 70 °C. While final chemical composition is difficult to estimate[62], overall density of the polymerized resin was $1.24 \pm 0.01$ g $ml^{-1}$ (mean ± s.e.m., $n = 15$ blocks of resin).

**Hard EPON resin protocols.** Resins 'A, B, C' fall within the broader description of 'hard EPON'[36]: an EPON812 or equivalent epoxide with hardeners MNA and a succinic anhydride at a weight ratio of 8:3:5 (ref. 36) (Supplementary Tables 2 and 3). Their polymerization is further catalyzed with BDMA. All reagents are viscous at room temperature. Resins 'A' and 'B' were prepared by mixing the epoxide and hardeners at room temperature in a closed container with a magnetic stirrer turning at gentle speed to avoid air being trapped in the viscous liquid. After the solution was homogeneous, the accelerator BDMA was added and stirred similarly until reaching a final homogeneous solution. For Resin 'C', each hardener was mixed with the epoxide separately first. Then, the two homogeneous mixtures of epoxy–hardener were stirred similarly as described above, in the presence of the accelerator BDMA (Supplementary Table 3).

**Resin D (TPTE) protocol.** For resin D, 2 g DDM was dissolved at 70 °C in 2 ml acetonitrile. Once the two components dissolved homogeneously, the solution was cooled down, 2 g of the epoxide TPTE (Supplementary Tables 2 and 3) was added and the same polymerization route as described below for resin E was followed.

**Resin E (TGPAP, 'tough') protocol.** After staining and dehydrating the samples as described above, the samples were immersed in 2× 100% acetonitrile (EMS-10020-450ML) for 30 min and 60 min, respectively. The TGPAP–DDM resin consists of the tri-functional epoxy resin TGPAP and the hardener DDM at a weight ratio of DDM:TGPAP = 1:2 (ref. 63)

(Supplementary Tables 2 and 3). As DDM does not mix well with TGPAP at room temperature and to facilitate the infiltration, DDM was first dissolved in acetonitrile heated to 70 °C and subsequently, TGPAP was added. The samples were incubated in 1:3 resin: acetonitrile for 2 h at room temperature, 1:1 resin: acetonitrile for 2–24 h at room temperature and subsequently the samples were placed in 1:1 resin: acetonitrile and cured for 12–72 h at 80 °C. As the boiling point of acetonitrile is at 82 °C, it is important to keep the sample container lid sufficiently open such that the acetonitrile can evaporate during the curing process (Supplementary Table 3).

**Sample verification.** To verify staining and sample integrity, all samples were imaged with a Zeiss Versa 510 laboratory-based micro-computed tomography[64]. The region of the sample to be targeted was defined based on the micro-computed tomography imaging results, and the sample was polished and trimmed to expose the layers of interest using a diamond knife (trim 90, Diatome).

**Pillar preparation.** A cylindrical sample was extracted from the resin-embedded tissue using a 30 keV Ga beam of 13 nA on a Zeiss NVision 40 Gallium FIB–SEM at the Paul Scherrer Institute (PSI) at the targeted locations. The integrated micromanipulator was used to mount the sample on the holder[65] for the PXCT measurement (a Ga-assisted C-depot was used for the fixation on the mount). The pillars were shaped and fine polished with a 30 keV Ga beam of 1.5 nA. The TGPAP pillars were first premilled with Preppy[66]. Fine polishing was performed with a Ga-FIB–SEM (TFS Helios 600 i, ScopeM) iteratively with currents decreasing from 65 nA to a final of 2.5 nA.

## PXCT
**Instrumentation.** PXCT was performed using the OMNY instrument[26] (Fig. 1a). The experiment was carried out at the cSAXS beamline of the SLS at the PSI, Villigen, Switzerland, during beamtimes e18628, e17964, e18604, e18536 and e19533 in 2019–2022 before shutdown for the SLS 2.0 multibend achromat construction. Details of the components are as follows. Coherent X-rays enter the instrument and pass optical elements that conform to an X-ray lens used to generate a coherent illumination onto the sample of a few µm. These elements are a gold central stop, a Fresnel zone plate (FZP) and an order-sorting aperture. The scattered X-rays are measured by a 2D detector (Eiger 1.5 M (ref. 67)). Accurate sample positioning is essential in X-ray ptychography and is achieved by horizontal and vertical interferometers that measure the relative position of the sample with respect to the FZP[28]. For this, the sample is directly mounted on a reference mirror, which is installed on a 3D piezo stage used for scanning the sample in the beam. A rotation stage allows recording projections at different orientations. All measurements were performed at a photon energy of 6.2 keV, corresponding to a wavelength of 2 Å, selected using a fixed-exit double-crystal Si(111) monochromator.

For datasets acquired in all beamtimes, except e19533, the FZP used, fabricated by the X-ray nano-optics group at PSI, had a diameter of 220 µm and 60 nm outermost zone width, resulting in a focal distance of 66.0 mm. For datasets acquired during beamtime e19533, we used a FZP from XRNanotech with 250 µm diameter and 30 nm outermost zone width, which had a focal distance of 37.5 mm. Both FZPs featured locally displaced zones, designed to produce an optimal structured illumination for ptychography[68], and resulted in a beam flux of about $7 \times 10^8$ photons s⁻¹. The beam transmitted through the FZP and the unwanted diffracted orders were blocked by the combination of a 40 µm diameter central stop and a 30 µm diameter order-sorting aperture. The FZP was coherently illuminated by using an upstream slit of 20 µm horizontal width, placed at about 12 m downstream of the undulator source, used as a secondary source. The detector was placed at about 7.2 m downstream of the sample.

**Data acquisition.** Samples were placed after the focal spot where the beam had a diameter of approximately 8 µm and 5 µm for the first four beamtimes and for the last beamtime e19533, respectively. In general, ptychographic scans were performed with a field of view that included the full sample size in the horizontal direction, while the scan points were positioned following a Fermat's spiral trajectory[69]. The dose on each 2D ptychographic projection was adjusted by combining different acquisition settings, namely (1) the average step size of the scan, ranging from 2.0 to 0.5 µm and (2) the exposure time at each scan point, ranging from 0.025 to 0.1 s. Reconstructions were performed from areas in the detector ranging between $480 \times 480$ pixels² and $700 \times 700$ pixels², resulting in reconstructed pixel sizes of between 41.7 nm and 27.6 nm.

Ptychographic reconstructions of the first four beamtimes were performed with a few hundred iterations of the difference map algorithm followed by a few hundred iterations of a maximum likelihood refinement[34]. Ptychographic reconstructions of the last beamtime e19533 were performed with two probe modes using 600 iterations of maximum likelihood.

For the tomography, several projections with equal angular spacing between sample rotations of 0° and 180° were recorded. The number of projections varied between approximately 300 and 1,200, providing another approach to adjust the dose imparted on each sample. The phase of the reconstructed projections was used after postprocessing alignment and removal of constant and linear-phase components[31,50]. A modified filtered back projection was used after aligning the projections using a tomographic consistency approach[50]. The tomograms were computed with a Hann or a RamLak filter for the first four beamtimes and for the last beamtime e19533, respectively. For nonrigid reconstruction, we employed the algorithm described in ref. 31.

**Dose estimation.** We estimated the dose as the total energy absorbed by the sample divided by the total mass of the sample. For this, the total number of photons absorbed were determined directly from the diffraction patterns, compared with measurements where the beam is not going through the sample. We relied on the linearity of the Eiger 1.5 M detector at maximum measured count rates of $10^5$ photons/(s × pixel) and on its close to 100% efficiency at the used photon energy of 6.2 keV. The sample mass was estimated from the measured volume and the total number of electrons in the sample, which is measured directly by PXCT. For the conversion from the total number of electrons to total mass we made an estimation of the sample composition based on a mixture in equilibrium with all the components used for the embedding resin and the staining, obtaining an average molecular mass estimation of about $1.9 \pm 0.1$ g mol⁻¹ (mean ± s.e.m.), where the error stems from the uncertainty in the sample composition. This estimation is reasonable if we compare it with the values calculated for lipids, 1.80 g mol⁻¹ (ref. 70), and for a mixture of resin and stain metals with five times higher metal content compared with the mixture in equilibrium, 1.99 g mol⁻¹. The latter scenario would cause an attenuation through the sample slightly larger than the one measured by X-ray ptychography, providing a reasonable upper limit for the metal content within the sample, and thus for the average molecular mass of the sample material. The error in the determination of the absorbed dose is mostly given by this uncertainty.

The two datasets highlighted in Fig. 5 depict the optimal results obtained when imaging at a dose of $3.8 \times 10^8$ Gy and $2.5 \times 10^9$ Gy, with sample volumes of about $20 \times 20 \times 20$ µm³ and $10 \times 10 \times 5$ µm³, respectively. The parameters employed in the acquisition of these two PXCT datasets in particular are detailed in Supplementary Table 4. A detailed description of all reconstructions is provided in ref. 60 and a summary can be found in Supplementary Table 1.

### FIB–SEM data acquisition
FIB–SEM was carried out using a Crossbeam 540 FIB–SEM with Atlas 5 for 3D tomography acquisition (Zeiss). The OMNY pin was coated with

a 10 nm layer of platinum, mounted horizontally on a standard 12.7 mm SEM stub using carbon cement (LeitC) and coated with a further 10 nm layer of platinum. This method of mounting the pin ensured the cylinder of tissue was positioned in free space and could be reoriented appropriately within the SEM for ion beam milling with consideration to the X-ray dataset. As the sample was cylindrical and tracking marks therefore could not be applied, direct tracking of slice thickness was not possible. Autofocus and autostigmation functions were carried out on an area close to the outer edge of the sample. Electron micrographs were acquired at 8 nm isotropic resolution, using dwell times of 12 µs (C319_EPL1), 6 µs with ×3 line averaging (Y357_3dot), or 8.5 µs (Y357_30um). During acquisition, the SEM was operated at an accelerating voltage of 1.5 kV with 1 nA current (C319_EPL1, Y357_30um) or 500 pA (Y357_3dot). The EsB detector was used with a grid voltage of 1,200 V. Ion beam milling was performed at an accelerating voltage of 30 kV and current of 700 pA. Approximate data acquisition times were 3 days 7 h and 3,260 slices (C319_EPL1), 2 days 3 h and 2,814 slices (Y357_3dot), and 3 days 18 h and 1,936 slices (Y357_30um).

The FIB–SEM dataset was later warped to the ptychography space using Bigwarp[71]. The warped data of the volume also imaged with ptychography were then exported to the ptychography dataset's space with a voxel size of 9.4 nm in $x,y,z$ in C319_EPL1 and 6.9 nm in Y357_3dot and in Y357_30mu (which is one-quarter of the voxel size of the native ptychography dataset, 37.6 nm and 27.6 nm, respectively). In this way, both datasets could be stored as different layers of a common dataset in webknossos[72]. This setup enabled quick toggling between imaging modalities at any particular location, and simplified the configuration of the synapse detection tasks.

### X-ray computed tomography at BM05 at ESRF

The resin B (EPON812) and resin E (TGPAP-embedded) samples shown in Fig. 2 and Supplementary Video 1 were fixed, stained and embedded as described in the previous sections. For X-ray computed tomography and continuous beam exposure they were mounted on standard aluminum microtomography holders. Measurements were done in identical conditions for the two samples. The samples were illuminated with a broad bandwidth beam with an average energy of 25 keV, filtered with 0.54 mm of Al. The detector used for imaging was composed of a 23-µm-thick $Lu_2SiO_5$ scintillator coupled to an infinity-corrected long-working-distance Mitutoyo objective (10×, NA 0.28) and a PCO Edge sCMOS camera. Tomographic scans were recorded with 3,000 projections over 360°, using an exposure time of 50 ms per frame. The pixel size was 0.73 µm. Each scan took about 3 min and 45 s and between two consecutive scans the sample was continuously exposed to the same beam without imaging for 20 min. The EPON-embedded sample was exposed for a total of 4 h, resulting in severe damage as can be observed in Fig. 2b. The TGPAP-embedded sample was exposed for an additional 1 h, thus a total of 5 h o, without generating any observable damage. The measurements were performed in air at room temperature. The detector used for imaging was composed of a 23-µm-thick $Lu_2SiO_5$ scintillator and a PCO Edge sCMOS camera.

### Data analysis

**FSC analysis.** To measure resolution, a custom implementation of the Fourier ring or shell correlation analysis was used. For the estimation of the 2D resolution of a projection image (Extended Data Fig. 2), we performed the 2D Fourier ring correlation (FRC) between that image and a second image acquired with identical experimental parameters. We then compared the FRC with a threshold using the 1-bit criterion, which is equivalent to a signal to noise ratio of 0.5 for each of the compared images[39]. This procedure provides an estimation of the half-pitch resolution for each of the individual images. For the estimation of the 3D resolution of a tomographic dataset, we computed the 3D FSC of two subtomograms, each computed using half of the tomographic projections. In this way, we obtained two datasets acquired independently,

albeit each with double angular sampling. We then compared the FSC with a threshold according to the ½-bit criterion, which corresponds to a signal to noise ratio of 0.4 for the full tomographic dataset.

**Synapse identification.** The synapse identification tasks were performed on the sample for which both FIB–SEM as well as ptychography data were available. Two synapse identification approaches are presented: a dendrite-centric one and a randomized 'captcha'-like detection. For the dendrite-centric synapse identification task, five dendrites evolving in straight trajectories in distinct directions were chosen and traced (in a skeleton format) from the ground-truth FIB–SEM dataset. The dendrites chosen all presented a pale cytoplasm, straight trajectory, no branches and consistent thickness of around 2 µm to ensure they could be followed in the datasets of both imaging modalities. Since the sample this dataset belongs to was extracted from the external plexiform layer of the olfactory bulb, these dendrites are likely to be lateral (and possibly apical) dendrites of projection neurons (mitral or tufted cells). Each dendrite skeleton was followed three times independently looking at one orthogonal plane only, and every time all features resembling synapses were annotated by seeding single nodes. This operation was performed for both the FIB–SEM as well as the ptychography dataset ($n$ = 725 initial nodes in FIB–SEM, 576 initial nodes in PXCT). All nodes seeded in the FIB–SEM dataset were assumed to be pointing to true synapses. On the basis of previous studies, synapse density was estimated to be of 1 2 synapse $µm^{-3}$ (ref. 73). First, we obtained a census of synapses by looking at the FIB–SEM data alone: nodes seeded ≤300 nm away from each other were assumed to be pointing to the same synapse, which led to a total census of 338 synapses in all 5 dendrites. Most synapses (273/338, 81%) were detected on the first pass in the FIB–SEM dataset, and only a small fraction of the final census was only found at the third and last pass (65/338, 19%), consistent with the assumption that synapses are well resolved in the FIB–SEM data. Next, the nodes seeded in the ptychography dataset were matched with the previously defined census of synapses through a similar process: FIB–SEM-validated synapse locations receiving a tag from the ptychography dataset within a 300 nm euclidean distance were marked as 'detected'. Most detected synapses in X-rays (148/225, 66%) were already detected on the first pass in the ptychography dataset, and only a small fraction were only detected at the second and last passes (77/225, 34%). A number of annotations of putative synapses in the ptychography datasets were not matched to any synapse. These annotations were further used to detect robust confounding factors when annotating synapses. Locations in the dataset receiving ≥2 tags within a ≤300 nm distance were defined as 'hotspots' for confusion. A total of 36 hotspots were detected, and their ultrastructure was revisited in the FIB–SEM dataset. Features providing the confusion were then identified by toggling the view of the ptychography and FIB–SEM dataset in the browser, and ultimately the nature of the feature leading to false positive detection in the ptychography dataset was annotated categorically. After revisiting the ultrastructure of all 36 false positive hotspots, we quantified the prevalence of the different feature categories.

For the 'captcha'-like synapse detection task, we generated 250 nonoverlapping and randomly located $1 \times 1 \times 1$ $µm^3$ regions of interest (Extended Data Fig. 8) in the best reconstruction of a 'low-dose' ($3.8 \times 10^8$ Gy) PXCT dataset. These regions could be displayed as cubes with three red faces and three green faces in the webknossos environment. The task consisted in determining whether a given region contained a synapse or not, by assigning a confidence score of 1, clearly not containing a synapse, or 2, 3 or 4, clearly containing a synapse. If the synapse was only partly contained inside the region, it would only be taken into account if it exits the region through a green boundary. In addition, ten locations known to contain a synapse (extracted from the previous dendrite-centric analysis) were chosen as training data. Finally, the task was encoded within the webknossos ecosystem so

an annotator was navigated from region to region after each answer was being recorded, and only one dataset (the one being tested each time) was presented at any point in time (Extended Data Fig. 8e,f). The task, overall, consisted of exploring the X-ray and FIB–SEM appearance of 10 training synapses (Extended Data Fig. 8b), assessing the presence of synapses in the 250 regions in the ptychography dataset (Extended Data Fig. 8c,e) and then assessing the presence of synapses in the 250 regions in the FIB–SEM dataset (Extended Data Fig. 8c,f). Three independent annotators, all experts in the appearance of synapses when their fine ultrastructure is resolved (for example, by FIB–SEM), ran the task. The annotator's expertise in synapse identification was monitored by their synapse detection scores in the FIB–SEM data, compared to the average of other two annotators (cross-correlation >40% required). The analyses returned therefore a 250 regions × 3 annotators × 2 imaging modalities array of responses, each with a value of either 1, 2, 3 or 4. In some cases, some annotators did not log a response before switching to the next task, which provided a value of 0. Only regions with correct assessments from all three annotators were kept, providing a cleaned up array of 240 × 3 × 2 responses. By matching region IDs, this array was represented in a 240 × 6 table and complemented with other regional metadata in additional columns. For every region, the average score from each imaging modality was calculated from all scores given by the three annotators. All four possible values were well represented in both imaging modalities. At this point, table rows were split into four groups according to the average FIB–SEM scores of each region (rounded to the closest integer) (Extended Data Fig. 8g). This allowed plotting the scores assigned in the ptychography data depending on the score given in the FIB–SEM data at the same region (Fig. 5d). The average responses of all regions in both modalities were later binned into two categories (average score <2.5 categorized as 'no synapse' or average score ≥2.5 categorized as 'synapse'). This allowed extracting a confusion matrix on the detectability of synapses (Extended Data Fig. 8h). This same analysis was then directed to the best reconstruction of a 'high-dose' ($2.5 \times 10^9$ Gy) dataset. The additional dose restricted the volume that could be imaged, and accordingly the volume analyzed, which could fit 30 nonoverlapping randomly allocated regions of interest (Extended Data Fig. 8d). A similar tracing task was submitted to 5 independent human annotators, one of whom was removed because their FIB–SEM-based synapse detection correlated <30% with the average of the other four annotators. After cleanup, this resulted in an array of 29 × 4 × 2 responses. An equivalent analysis followed the one described above, resulting in 4 × 4 confusion matrices (Extended Data Fig. 8i) that provided the data to generate the plot in Fig. 5i, and that were then condensed into 2 × 2 confusion matrices indicating the detection of 'synapse' or 'no synapse' (Extended Data Fig. 8j). The 2 × 2 confusion matrices provided the scores of true positives (TP), false positives (FP) and false negatives (FN) that were used to calculate the precision (*P*), recall (*R*), *F*1 and *F*$_\beta$ detection scores for synapse detection

$P = \text{TP}/(\text{TP} + \text{FP})$

$R = \text{TP}/(\text{TP} + \text{FN})$

$F1 = 2 \times P \times R/(P + R)$

$F_\beta = (1 + \beta^2) \times P \times R/((\beta^2 \times P) + R); \beta = 2$, such that recall is weighted higher than precision.

**Segmentation.** The manual segmentation and renderings shown in Figs. 1c and 6c and Supplementary Video 3 were generated using the image analysis services of https://ariadne.ai.

Segmentations shown in Extended Data Fig. 9 were generated as follows: pre-existing segmentation models by zetta.ai were used for segmenting both the PXCT as well as the corresponding FIB–SEM datasets without further fine tuning or proofreading. The PXCT image intensity range was stretched, so that values of 70 mapped to 0 and 120 mapped to 255. An affinity-based segmentation pipeline[74,75] was applied using a pre-existing segmentation model that had been trained to produce affinities and dense segmentation using a mouse cerebellum data collected with FIB–SEM, where the input and output resolution were 16 nm isotropic. The model was applied to an isotropic 14 nm downsampled FIB–SEM dataset and to the PXCT data at native 28 nm isotropic voxel size. Various watershed and agglomeration parameters were tested and manually inspected to identify a reasonable best-effort segmentation.

**Software.** Data processing to reconstruct the X-ray ptychographic reconstructions was performed using custom beamline code in MATLAB at the cSAXS beamline. Datasets acquired at BM05 in ESRF were reconstructed using Nabu (https://tomotools.gitlab-pages.esrf.fr/nabu/about.html). Data analysis was performed using MATLAB (R2022a-R2024b). Beamline diagrams were designed with Catia V5 and CorelDRAW (version 25.1.0.269). The 3D renders were generated with Blender (www.Blender.org).

### Reporting summary

Further information on research design is available in the Nature Portfolio Reporting Summary linked to this article.

### Data availability

Source data of the graphs presented in the main figures are provided in ref. 60. All 3D datasets and major annotations reported in this study are accessible through the associated code repository (see 'Code availability'). A supplementary structured table including metadata supporting the measurements presented is available via Zenodo at https://doi.org/10.5281/zenodo.16362800 (ref. 60). The dense segmentation of both datasets can be found in ref. 61 and under https://spelunker.cave-explorer.org/#!middleauth+https://global.daf-apis.com/nglstate/api/v1/5950833582669824.

### Code availability

Analysis code is available (under MIT license) via GitHub at https://github.com/cboschp/ptychoStainedTissue (ref. 61). The ptychography reconstruction code is available from https://www.psi.ch/en/sls/csaxs/software (license: https://www.psi.ch/sites/default/files/import/sls/csaxs/ComputingEN/License.txt). All datasets can be viewed, annotated and downloaded from the links provided in Supplementary Table 1, also accessible through the aforementioned code repository on GitHub[61].

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

## Acknowledgements

We are grateful to the biological research and scientific computing science technology platforms of the Francis Crick Institute and to the electron microscopy facility at PSI. We thank J. Waters, T. Rees and J. Hess for insights on resin monomers and R. Roberts for expert annotation of synapses. We acknowledge the Paul Scherrer Institute, Villigen, Switzerland, and the ESRF, Grenoble, France, for the provision of synchrotron radiation beamtime at the cSAXS beamline of the SLS (proposals 20190654 to A.D., C.B. and A.P.; 20200783 to C.B., A.T.S., A.P. and A.D.; and 20211852 to A.A.W., C.B., A.T.S., A.D. and A.P.) and at the BM05 beamline of the European Synchrotron (proposals IH-LS-3497and IH-LS-3529 to A.P.), respectively. We gratefully acknowledge ScopeM at ETHZ and, in particular, J. Reuteler for their support and assistance in this work. For the purpose of open access, the author has applied a CC BY public copyright licence to any Author Accepted Manuscript version arising from this submission. This work was supported by the Francis Crick Institute, which receives its core funding from Cancer Research UK (grant nos. CC2036 to A.T.S. and CC1076 to L.C.), the UK Medical Research Council (grant nos. CC2036 to A.T.S. and CC1076 to L.C.) and the Wellcome Trust (grant nos. CC2036 and 110174/Z/15/Z to A.T.S. and CC1076 to L.C.). It was also supported by a Physics of Life grant (EP/W024292/1) to A.T.S. and A.P. funded by EPSRC and Wellcome. A.P. acknowledges funding from the European Research Council under the European Union's Horizon 2020 Research and Innovation Programme (no. 852455). A.A.W. acknowledges funding from the SERI-funded ERCSt MB22.00042. The work of T.A. is supported by funding from the Swiss National Science Foundation (SNF), project number 200021_196898.

## Author contributions

Conceptualization by C.B., A.D., A.A.W. and A.T.S. Sample preparation by C.B., E.M., Y.Z. and A.A.W. Algorithm development by T.A., M.G.-S. and M.H. Endstation development by T.A., M.H., O.B., A.M., M.G.-S., G.A. and A.D. PXCT data acquisition by C.B., T.A., M.H., A.P., M.G.-S., A.D., A.A.W. and A.T.S. SXRT data acquisition by A.P. and P.C. FIB–SEM data acquisition by C.J.P. and L.C. Data analysis by C.B. and T.A. Manuscript first draft by C.B. and A.T.S. Figure preparation by C.B., A.P., A.D. and A.A.W. All authors contributed to the editing of the paper.

## Funding

## Competing interests

A.A.W. is founder and owner of ariadne.ai ag. The other authors declare no competing interests.

## Additional information

**Extended data** is available for this paper at https://doi.org/10.1038/s41592-025-02891-0.

**Correspondence and requests for materials** should be addressed to Ana Diaz, Adrian A. Wanner or Andreas T. Schaefer.

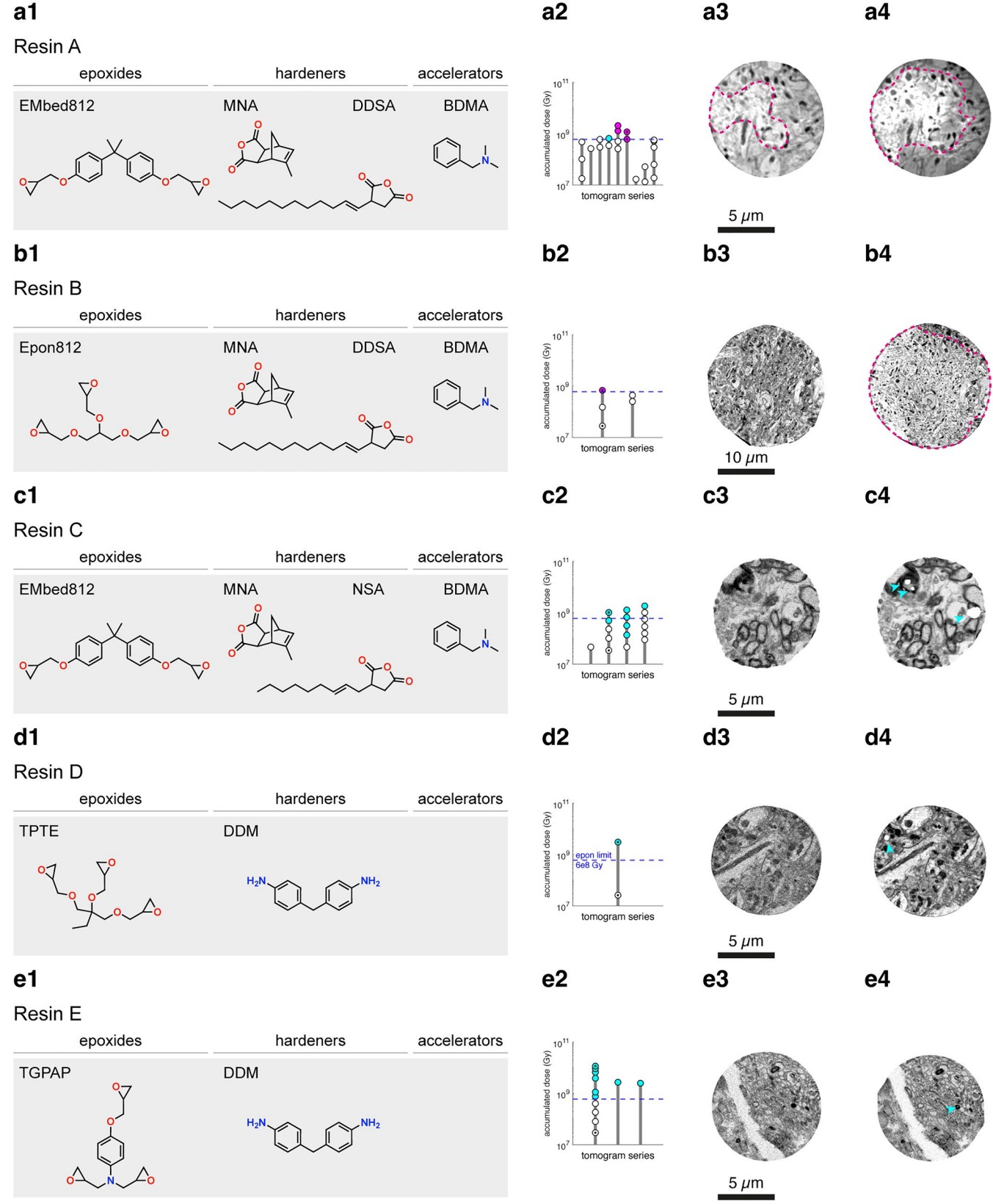

**Extended Data Fig. 1 | Epoxy resins. (a1)** Monomers employed in the formulation of resin A. **(a2)** Tomogram series acquired with each resin, showing stacked doses of all tomograms for each series, allowing to read in the y-axis the total accumulated dose absorbed by the sample at the end of each tomogram. Colour coding of tomograms is as in Extended Data Fig. 5e,j, with tomograms displaying localised compact mass loss labelled in cyan, and those displaying widespread mass loss incompatible with high-resolution reconstruction coloured in magenta. **(a3-4)** Representative cross-section of the reconstructed first (a3) and last (a4) tomogram from a series of tomograms obtained from the same sample location. These tomograms are indicated in (a2) with a black dot. Widespread (magenta) mass loss resulting from extended irradiation is indicated. **(b-e)** Same as in (**a**) for the epoxy resins B-E, respectively, used to embed the mouse brain tissue samples reported in this study.

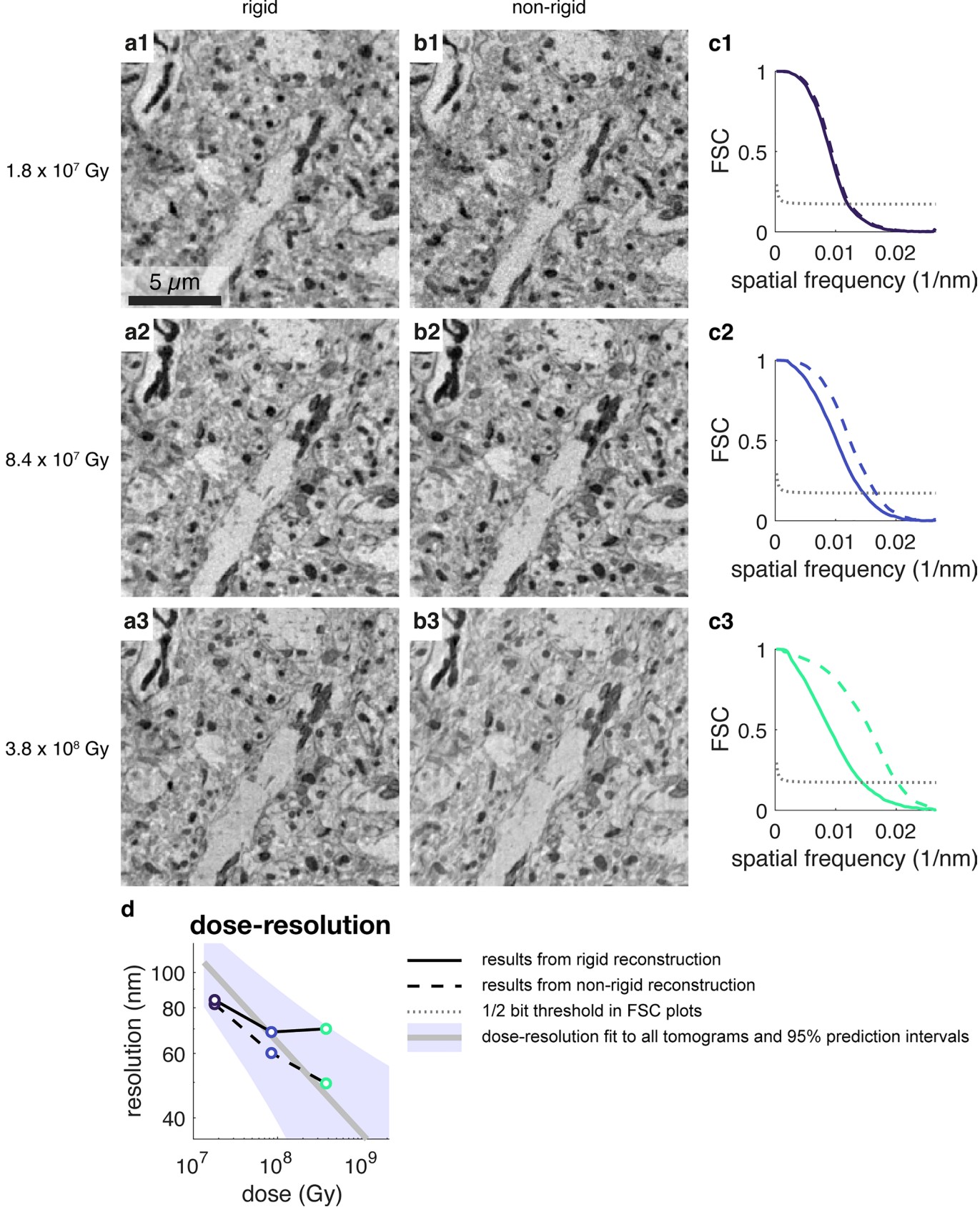

**Extended Data Fig. 2 | See next page for caption.**

**Extended Data Fig. 2 | Dose-dependent resolution of the reconstructed tomograms.** (**a1-a3**) Example images showing approximately the same region imaged at distinct doses, after a rigid tomographic reconstruction of all subtomograms involved in each case, for a mouse brain EPL sample embedded in resin A (EMbed812). (**b1-b3**) Same images shown in a1-a3 after non-rigid tomographic reconstruction of the same data. (**c1-c3**) Fourier shell correlation (FSC) for the three tomograms shown in a,b and both reconstruction algorithms (solid line: rigid tomographic reconstruction; dashed line: non-rigid tomographic reconstruction). (**d**) Resolution estimated by FSC in c1-c3 as a function of absorbed dose. Solid and dashed lines indicate rigid and non-rigid tomographic reconstruction, respectively. The solid grey line indicates the fit *FSC_resolution = a/dose^(¼)* for all datasets as described in Fig. 2d, along the 95% prediction intervals with observation, non-simultaneous bounds (purple shaded area). [Resolution in the non-rigid reconstruction was different than in the rigid reconstruction by more than 1 nm in 28/59 tomograms, and in all those 28/28 cases resolution was improved in the non-rigid reconstruction].

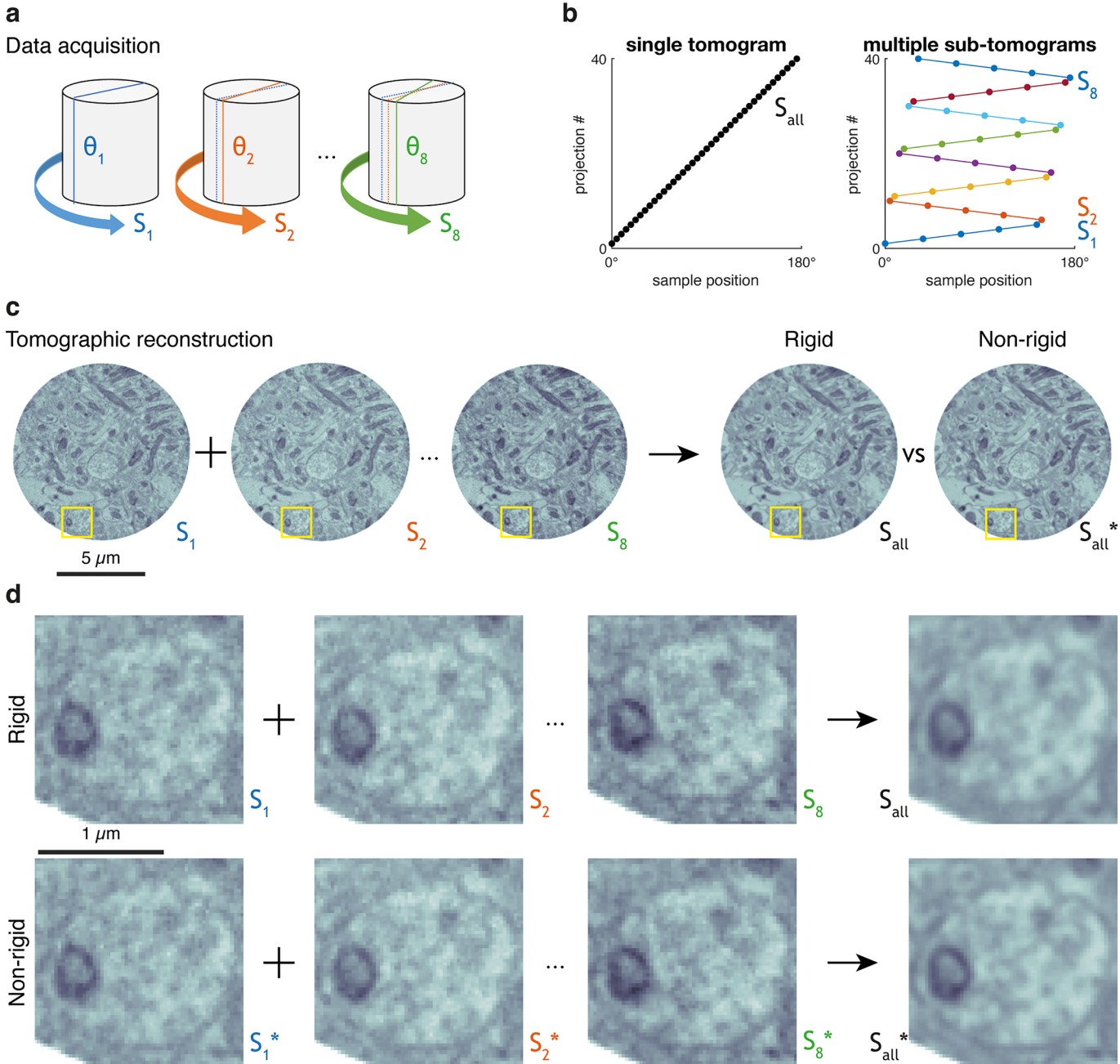

**Extended Data Fig. 3 | Subtomogram acquisition and non-rigid tomogram reconstruction.** (a-b) Schematics of a subtomogram acquisition for a mouse brain EPL sample embedded in resin A (EMbed812). Data are acquired in multiple tomograms, each sampling the same angle but starting with an offset from each other (**a**). This allows for an acquisition algorithm where all desired angles are sampled but in distinct subtomograms (**b**), each of them enabling independent reconstruction. (**c-d**) Reconstructed subtomograms can later be summed to generate the final reconstruction. This sum can be applied either directly (through a 'rigid' approach of all subtomograms $S_1 \ldots S_8$) or after warping all subtomograms to a common space ('nonrigid' sum of all warped tomograms $S_1^* \ldots S_8^*$). Any sample deformations occurring during acquisition will affect a limited subset of subtomograms, and their effect of worsening reconstruction quality is then minimised by the nonrigid reconstruction approach (see insets highlighted in c magnified in d for both reconstruction modes).

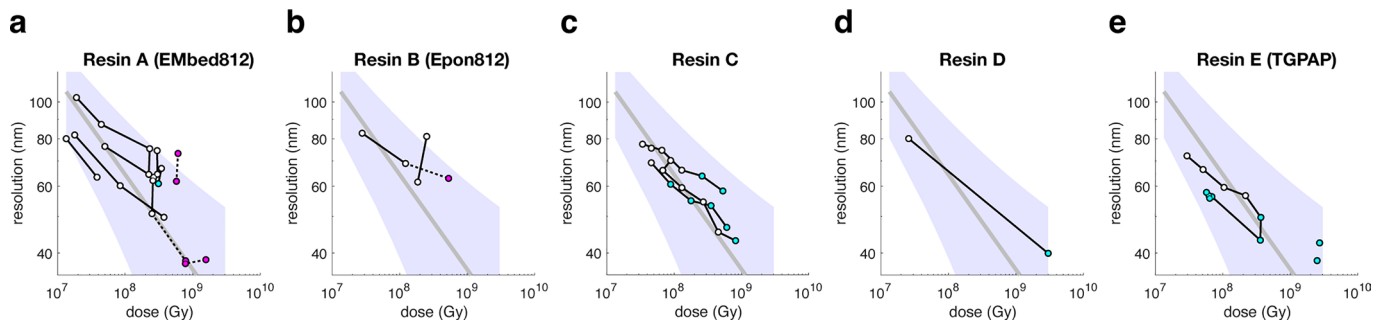

**Extended Data Fig. 4 | Dose-dependent resolution for samples embedded in all resins.** (**a-e**) Fourier shell correlation (FSC) resolution for all tomograms of all samples reported, imaged at different doses, as shown in Fig. 3d, split in different subplots by the resin each sample was embedded in. Tomograms (circles) obtained of the same sample at multiple doses form a series and are connected with a black line. The color of the circle indicates whether the tomogram had no radiation-induced damage (white), had contained, globular and compact ( < 100 nm) low-absorbing regions compatible with reconstruction and analysis (cyan) or a diffuse, sparse mass loss (magenta). Examples of the different radiation damage effects are shown in Fig. 2 and in Extended Data Fig. 2. The power function *FSC_resolution = a/dose^(¼)* (grey line) fitted to all white and cyan data points from all resins pooled together along its 95% prediction intervals with observation, non-simultaneous bounds (purple shaded area) is shown in all plots.

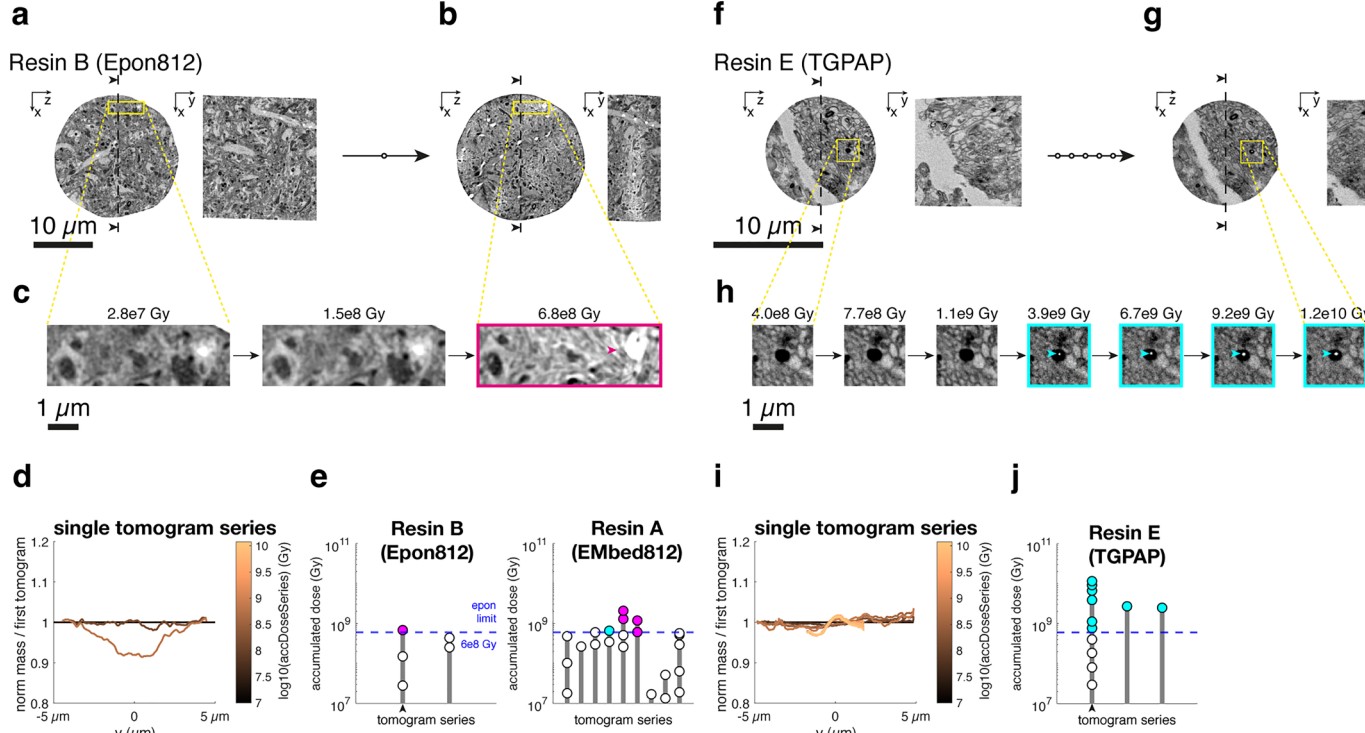

**Extended Data Fig. 5 | Widespread vs localized compact mass loss defects.**
(**a-c**) Radiation stress tests performed at cryogenic temperatures under vacuum with a 6.2 keV monochromatic X-ray beam (PSI cSAXS) using ptychographic X-ray tomography on similar samples embedded with resin B (Epon812). The same tissue region was imaged multiple times successively (a-b) providing a series of tomograms that allowed for both assessing the sample structure and monitoring the radiation dose absorbed. (**d**) Mass loss across the vertical midline (y axis) of the sample shown in (a-c) across successive tomograms.(**e**) Samples embedded in resins A and B (EMbed812 and Epon812) consistently displayed signs of widespread mass loss (magenta) after absorbing $6.8×10^8$ Gy [n = 0/10 tomogram series displayed widespread mass loss when imaged with an accumulated dose of $<6×10^8$ Gy; n = 3/4 tomogram series displayed this pattern when imaged with an

accumulated dose of $>6×10^8$ Gy, with the remaining tomogram series displaying constrained damage at $6.56×10^8$ Gy]. (**f-h**) Radiation stress tests as described in (**a-c**), performed on a sample embedded in resin E (TGPAP). Spatially constrained radiation damage ( ~ 100 nm diameter) was observed in samples irradiated with cumulative doses above $6.8×10^8$ Gy (cyan). (**i**) Mass across the sample midline is conserved across successive tomograms. (**j**) Samples embedded in resin E (TGPAP) did not display signs of mass loss across successive tomograms, withstanding accumulated radiation doses above $1.15×10^{10}$ Gy [n = 0/1 tomogram series displayed widespread mass loss when imaged with an accumulated dose of $<6×10^8$ Gy; n = 0/3 tomogram series showed it either when imaged with accumulated doses of $≥6×10^8$ Gy].

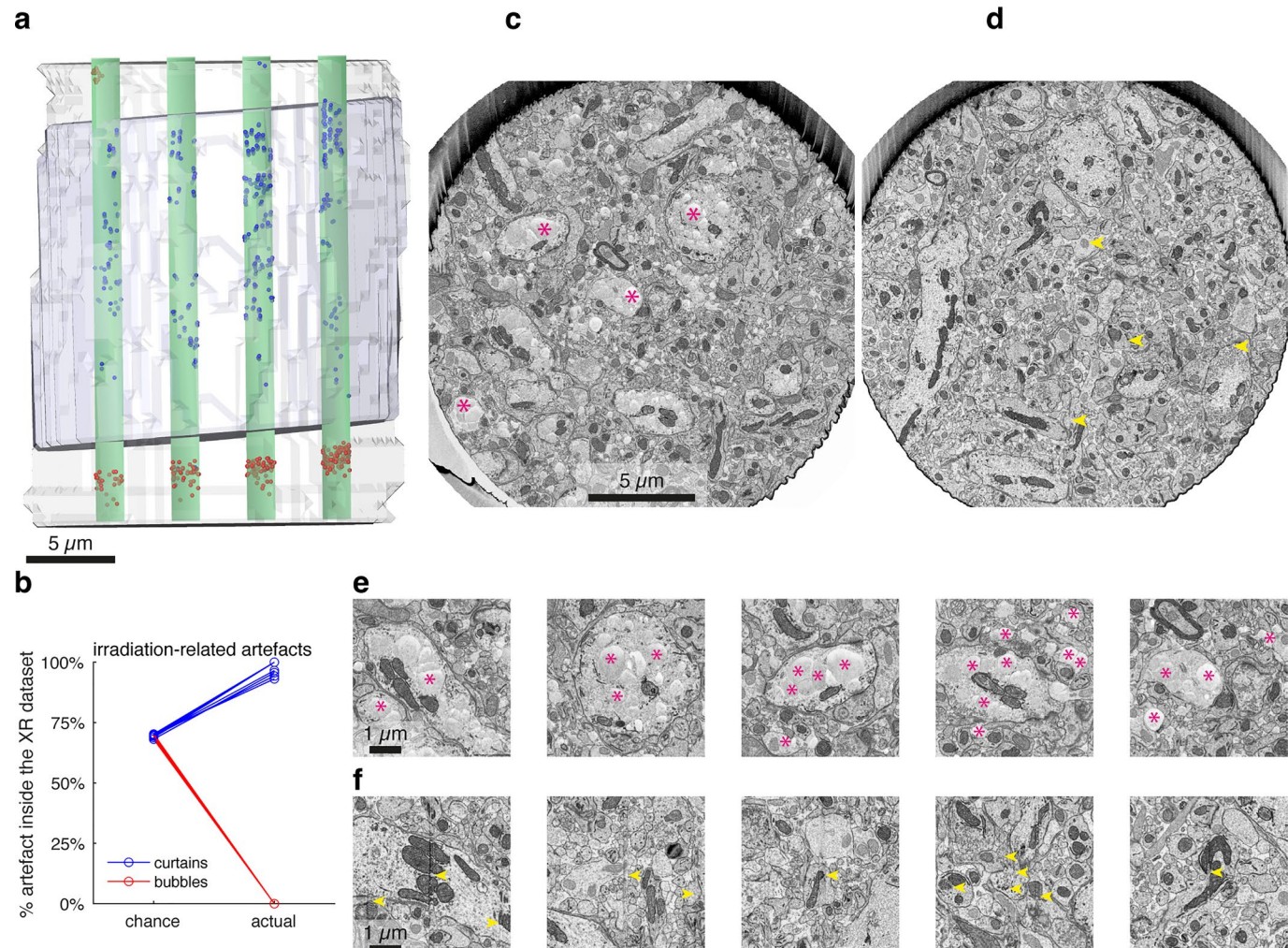

**Extended Data Fig. 6 | FIB-SEM investigation of ultrastructure. (a)** 3D render of a mouse brain EPL sample embedded in resin A (EMbed812) that was imaged with both PXCT (blue mesh) and FIB-SEM (gray mesh). The FIB-SEM dataset fully contained the region imaged with PXCT. A 4x4 grid of column ROIs (green, all annotations but only 4 columns visible from the chosen perspective) was defined and sample defects of two categories ('curtains' in FIB-SEM imaging, blue dots and 'bubbles', red dots) were annotated throughout the ROIs. For each ROI, its overlap between the two datasets was calculated, and later used to assess the likelihood of artifact distribution by chance vs influenced by X-ray imaging. **(b)** Distribution of the two artifact types inside and outside of the PXCT-imaged region. Chance level shows, for each ROI, the expected share of artifacts that should have been found in the PXCT-imaged region. Actual level shows the % of curtains (blue) and bubbles (red) found inside the PXCT-imaged region in each ROI. [This analysis was performed on the n = 1 FIBSEM dataset obtained from samples embedded in resins A (EMbed812) or B (Epon812) and imaged with PXCT. n = 2 additional samples embedded in resin E (TGPAP) and imaged with PXCT were also imaged later with FIBSEM. Of these, n = 0/2 presented bubbles at the region adjacent to the irradiated zone; n = 1/2 did not show any curtaining while the other presented curtaining throughout both irradiated and non-irradiated areas.]. **(c-d)** Cross-section of the FIB-SEM dataset at two heights, showing examples of the artifacts 'bubble' (c, magenta asterisks) and 'curtain' (d, yellow arrowheads). **(e-f)** Close-up views of the artefacts shown in c and d, respectively.

**a**                                                **b**

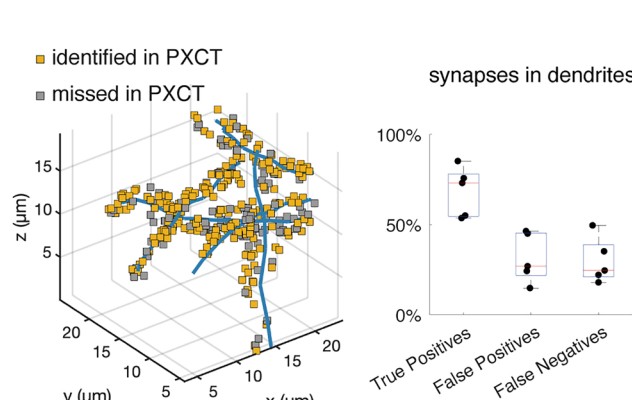

**Extended Data Fig. 7 | Synapse detection in targeted dendrites in PXCT.**
(**a**) Centerline of 5 dendrites and locations of all afferent synapses into these dendrites, tagged in both the PXCT (sample embedded in resin A (EMbed812), dose 3.8x10$^8$ Gy) and FIB-SEM datasets (same datasets as in Fig. 5a,b). The ground truth census of synapses was extracted from the consensus identification in the FIB-SEM dataset (all squares, regardless of colour). Tagged putative synapses in the PXCT dataset within 300 nm of a consensus synapse location were considered a positive detection of a synapse in PXCT. Color-coding shows synapses detected in both datasets (orange squares, n = 225) or only in the FIB-SEM dataset (grey squares, n = 113). (**b**) Synapse detection scores, grouping results by dendrite (n = 5 dendrites).

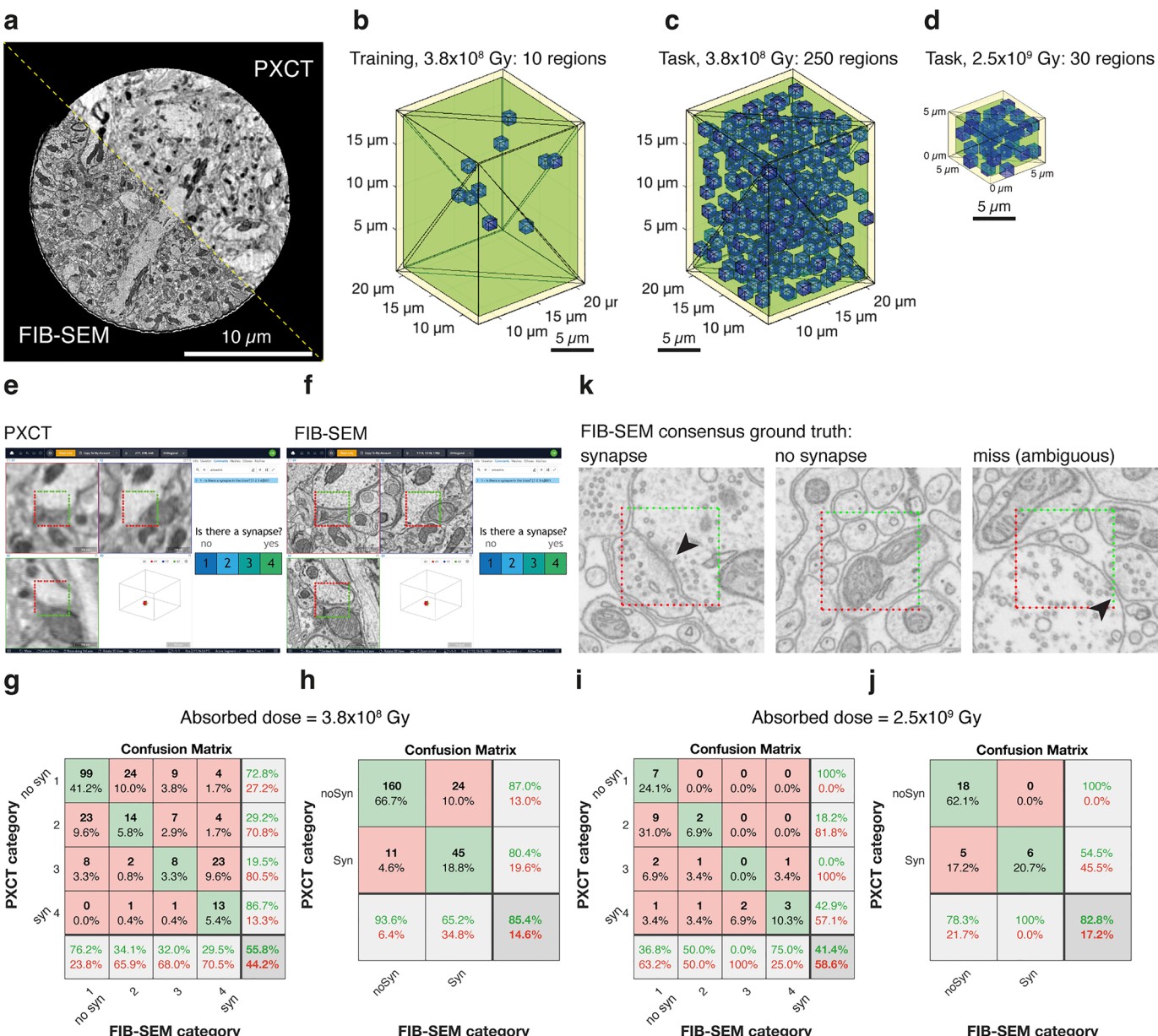

**Extended Data Fig. 8 | Synapse detection in PXCT. (a)** Cross-section of a sample imaged by both PXCT and FIB-SEM, showing the same plane viewed by either imaging modality. **(b)** Spatial distribution of 10 non-overlapping regions of interest in the $3.8 \times 10^8$ Gy PXCT dataset's space coordinates, each region occupying 1 µm³ (blue), used for training before the synapse identification task. **(c)** Spatial distribution of 250 non-overlapping regions of interest in the $3.8 \times 10^8$ Gy PXCT dataset's space coordinates, used for the synapse identification task. **(d)** Spatial distribution of 30 non-overlapping regions of interest in the $2.5 \times 10^9$ Gy PXCT dataset's space coordinates, used for the synapse identification task. **(e-f)** Synapse 'captcha' detection task: a human annotator is presented with each region of interest, displaying either the PXCT **(e)** or FIB-SEM **(f)** volume, and must annotate whether the region contains a synapse (score = 4) or it doesn't (score=1). All responses could be obtained from 3 human tracers across

240 regions of interest in the $3.8 \times 10^8$ Gy PXCT dataset and its correlative FIB-SEM, and across 29 regions of interest from 4 human tracers in the $2.5 \times 10^9$ Gy PXCT/FIB-SEM experiment. Score values distributed equally regardless of imaging modality. **(g-h)** Confusion matrices of synapse detectability in PXCT using FIB-SEM as ground truth, showing all 4 categories **(g)** and pooling the results into 2 categories ('synapse' vs 'no_synapse') **(h)**, for the $3.8 \times 10^8$ Gy PXCT experiment. **(i-j)** Same as in g-h, for the $2.5 \times 10^9$ Gy PXCT experiment. For $3 \times 10^8$ Gy, recall was 65% and precision 80%; for $2.5 \times 10^9$ Gy (with a smaller sample size) recall increased to 100%. Precision dropped to 55%. The calculation of "precision" and "recall" assumes a "perfect" ground truth FIB-SEM synapse assignment. **(k)** shows an example of a synapse detected only by 2 out of 4 annotators in the FIBSEM dataset (missed by 2/4) that was detected by 3/4 in the $2.5 \times 10^9$ Gy PXCT dataset.

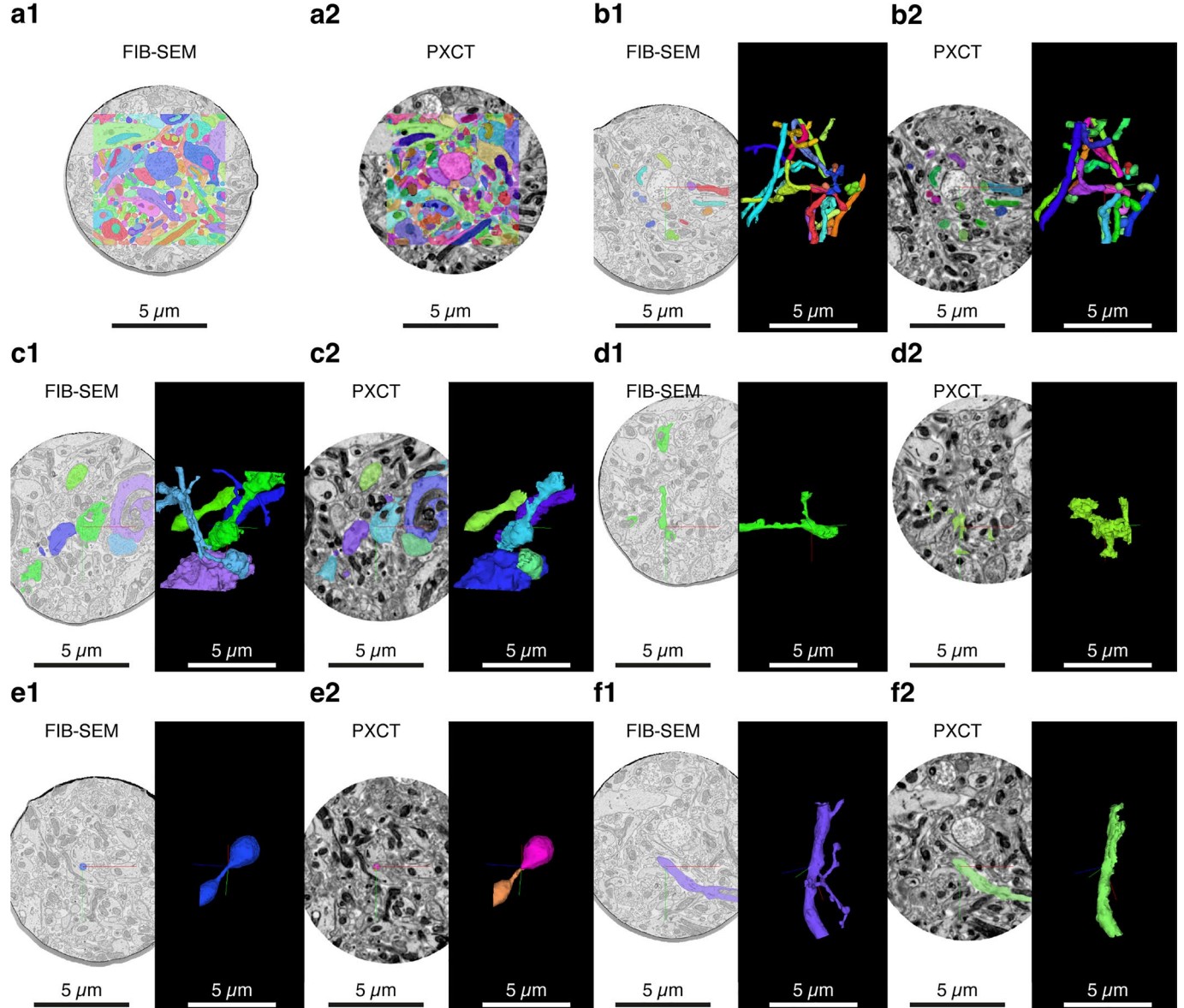

**Extended Data Fig. 9 | Automated segmentation of biological features in PXCT.**
(**a**) Automated dense segmentation of all biological features in the best efforts PXCT dataset (2.5x10⁹ Gy, TGPAP-embedded) using a state of the art classifier trained on FIBSEM data[75]. Segmentation results are shown for the classification of PXCT data (**a2**) as well for the same region imaged with FIBSEM (**a1**).

(**b**) A selection of mitochondria, automatically segmented in both FIBSEM (**b1**) and PXCT (**b2**). (**c**) A selection of neurites, automatically segmented in both FIBSEM (**c1**) and PXCT (**c2**). (**d-f**) Automated neurite segmentation returns errors of both types: mergers (**d**, multiple neurites merged into a single object) and splits (**e-f**, a single neurite split into multiple objects).

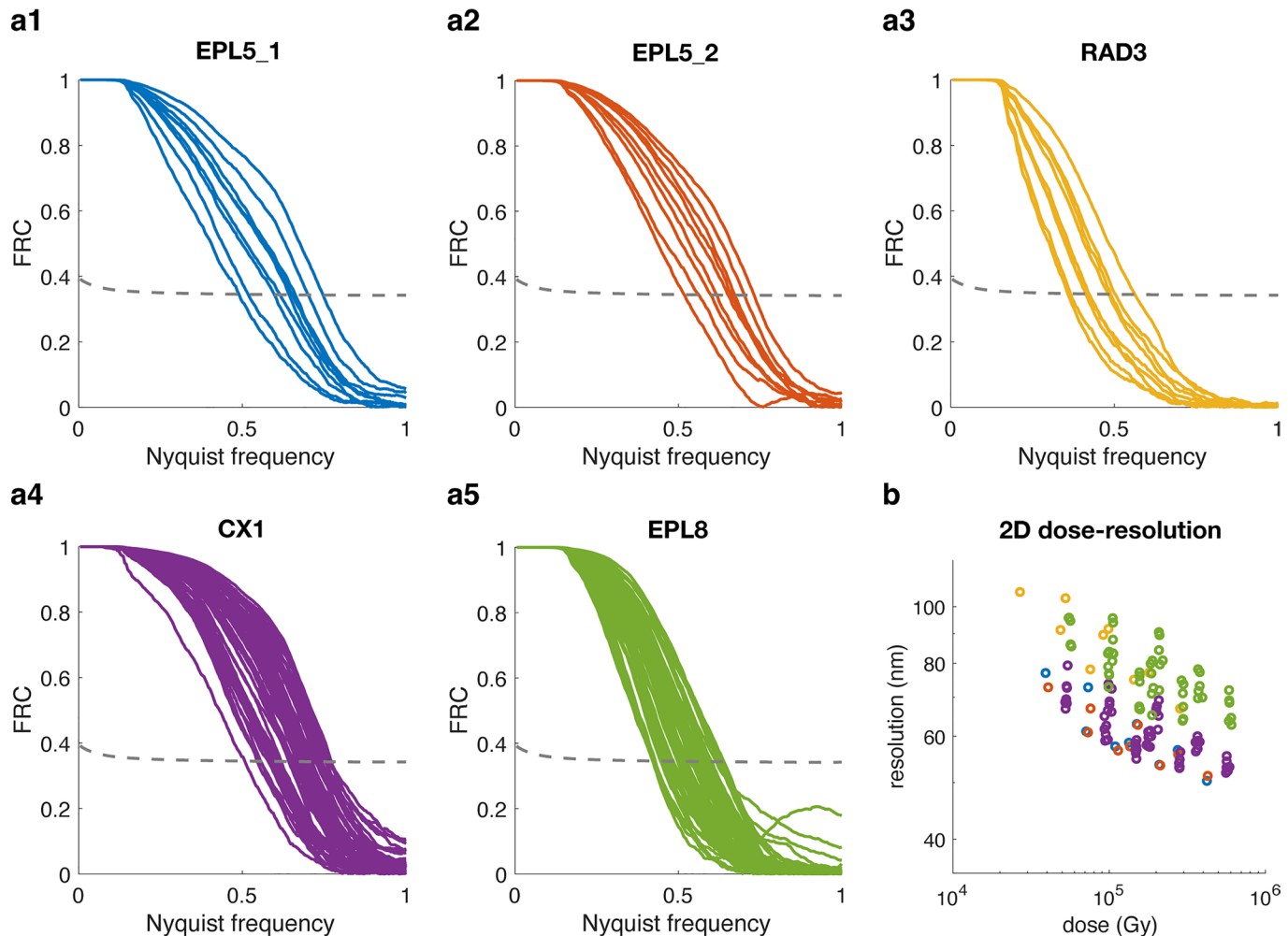

**Extended Data Fig. 10 | Dose-dependent resolution measurements.**
(**a1-a5**) Fourier Ring Correlation (FRC) obtained from 2D PXCT images obtained across different X-ray doses on multiple samples from three mouse brain tissue regions (EPL, RAD, CX) embedded in resin A (EMBed812) (EPL5_1, EPL5_2, RAD3, CX1, EPL8) and in resin B (Epon812) (EPL8). (**b**) Resolution estimates for all scans in a1-a5. [n = 5 2D PXCT experiments plotted, color coded as in panels a1-a5. Each experiment provides one datapoint per replicate at 9 different doses. Experiments EPL5_1, EPL5_2 and RAD3 contained a single replicate (n = 9 data points each), whereas experiments CX1 and EPL8 contained 7 (n = 63 data points) and 6 replicates (n = 54 data points), respectively].

Ana Diaz
Adrian A. Wanner

# Reporting Summary

## Statistics

For all statistical analyses, confirm that the following items are present in the figure legend, table legend, main text, or Methods section.

| n/a | Confirmed | |
|-----|-----------|---|
| ☐ | ☒ | The exact sample size (*n*) for each experimental group/condition, given as a discrete number and unit of measurement |
| ☐ | ☒ | A statement on whether measurements were taken from distinct samples or whether the same sample was measured repeatedly |
| ☒ | ☐ | The statistical test(s) used AND whether they are one- or two-sided <br> *Only common tests should be described solely by name; describe more complex techniques in the Methods section.* |
| ☒ | ☐ | A description of all covariates tested |
| ☒ | ☐ | A description of any assumptions or corrections, such as tests of normality and adjustment for multiple comparisons |
| ☐ | ☒ | A full description of the statistical parameters including central tendency (e.g. means) or other basic estimates (e.g. regression coefficient) AND variation (e.g. standard deviation) or associated estimates of uncertainty (e.g. confidence intervals) |
| ☒ | ☐ | For null hypothesis testing, the test statistic (e.g. $F$, $t$, $r$) with confidence intervals, effect sizes, degrees of freedom and $P$ value noted <br> *Give P values as exact values whenever suitable.* |
| ☒ | ☐ | For Bayesian analysis, information on the choice of priors and Markov chain Monte Carlo settings |
| ☒ | ☐ | For hierarchical and complex designs, identification of the appropriate level for tests and full reporting of outcomes |
| ☒ | ☐ | Estimates of effect sizes (e.g. Cohen's *d*, Pearson's *r*), indicating how they were calculated |

*Our web collection on statistics for biologists contains articles on many of the points above.*

## Software and code

Policy information about availability of computer code

| | |
|---|---|
| Data collection | Most datasets reported in this manuscript were acquired at the cSAXS beamline of the Paul Scherrer Institut (Swiss Light Source). Scattered X-rays were measured by a 2D detector (Eiger 1.5M). For the datasets obtained at BM05 (European Synchrotron), The detector used for imaging was composed of a 23 µm thick LSO scintillator coupled to an infinity-corrected long-working-714 distance Mitutoyo objective (10x, NA 0.28) and a PCO Edge sCMOS camera. |
| Data analysis | Data processing to reconstruct the X-ray ptychographic reconstructions was performed using custom beamline code in MATLAB at the cSAXS beamline. Datasets acquired at BM05 in ESRF were reconstructed using Nabu (https://tomotools.gitlab-pages.esrf.fr/nabu/about.html). Data analysis was performed using MATLAB (R2022a-R2024b, 9.12.0.1884302). Beamline diagrams were designed with Catia V5 and CorelDRAW (version 25.1.0.269, June 2024 Release). 3D renders were generated with Blender (www.Blender.org; Blender Online Community. (2021). Blender - A 3D modelling and rendering package. Blender Foundation, Stichting Blender Foundation, Amsterdam.) <br><br> Supporting code: <br> https://github.com/cboschp/ptychoStainedTissue <br> Non-destructive X-ray tomography of brain tissue ultrastructure. Supporting code ptychoStainedTissue. (v1.1.0). Zenodo. https://doi.org/10.5281/zenodo.16364654 |

For manuscripts utilizing custom algorithms or software that are central to the research but not yet described in published literature, software must be made available to editors and reviewers. We strongly encourage code deposition in a community repository (e.g. GitHub). See the Nature Portfolio guidelines for submitting code & software for further information.

## Data

Policy information about availability of data

All manuscripts must include a data availability statement. This statement should provide the following information, where applicable:

- Accession codes, unique identifiers, or web links for publicly available datasets
- A description of any restrictions on data availability
- For clinical datasets or third party data, please ensure that the statement adheres to our policy

Source data of the graphs presented in the main figures are provided as a Source Data file. The datasets and major annotations reported in this study are accessible through the associated code repository (see Code Availability).

Supporting dataset:
Non-destructive X-ray tomography of brain tissue ultrastructure. Supporting metadata. [Data set]. Zenodo. https://doi.org/10.5281/zenodo.16362800

## Human research participants

Policy information about studies involving human research participants and Sex and Gender in Research.

| Reporting on sex and gender | This study did not involve human participants. |
| Population characteristics | See above. |
| Recruitment | n/a |
| Ethics oversight | n/a |

Note that full information on the approval of the study protocol must also be provided in the manuscript.

# Field-specific reporting

Please select the one below that is the best fit for your research. If you are not sure, read the appropriate sections before making your selection.

☒ Life sciences          ☐ Behavioural & social sciences          ☐ Ecological, evolutionary & environmental sciences

For a reference copy of the document with all sections, see nature.com/documents/nr-reporting-summary-flat.pdf

# Life sciences study design

All studies must disclose on these points even when the disclosure is negative.

| Sample size | Tomogram series were defined as a series of consecutive tomograms obtained from the same specimen imaging the same field of view (ie without >2μm variations in sample height) and without any unaccounted exposures that would irradiate >1e6 Gy. This allowed to monitor the accumulated X-ray dose absorbed by the sample across the series. Cylindrical samples of 5-30 μm in width were prepared from mouse brain tissue, targeting whenever possible tissue regions devoid of uninformative 10μm+ large features such as blood vessels or cell nuclei. The external plexiform layer of the olfactory bulb provided a reproducible background within and across individuals, containing a broad range of biological feature sizes (10-1000nm). Sample size was ultimately limited by acquisition (available beamtime) and preparation (FIB availability). |
| Data exclusions | No datasets were excluded from the analysis.<br>In the synapse detection task, only data series arising from annotators that completed the whole task were kept (n=3 annotators). Within those, cube locations that were missed by any of the three annotators were discarded, leading to a complete response set for 240 locations by 3 annotators that was further analysed. |
| Replication | We report results arising from 63 tomograms organised in 19 tomogram series acquired from 17 separate samples during 5 beamtimes. Special care was addressed to extend the amount of technical and biological replicates whenever possible. |
| Randomization | All tomograms were analysed in bulk. |
| Blinding | Data analysis of pooled tomogram series can be traced to each individual tomogram whenever possible, and individual data points are always shown. |

# Reporting for specific materials, systems and methods

We require information from authors about some types of materials, experimental systems and methods used in many studies. Here, indicate whether each material, system or method listed is relevant to your study. If you are not sure if a list item applies to your research, read the appropriate section before selecting a response.

## Materials & experimental systems

| n/a | Involved in the study |
|---|---|
| ☒ | Antibodies |
| ☒ | Eukaryotic cell lines |
| ☒ | Palaeontology and archaeology |
| ☐ ☒ | Animals and other organisms |
| ☒ | Clinical data |
| ☒ | Dual use research of concern |

## Methods

| n/a | Involved in the study |
|---|---|
| ☒ | ChIP-seq |
| ☒ | Flow cytometry |
| ☒ | MRI-based neuroimaging |

# Animals and other research organisms

Policy information about studies involving animals; ARRIVE guidelines recommended for reporting animal research, and Sex and Gender in Research

| | |
|---|---|
| Laboratory animals | Animals used in this study were 6-16 week old wildtype mice of C57Bl/6 and CD-1 background of either sex. Mice were housed up to 5 per cage with an ad libitum supply of food and water under a 12–12 h light–dark cycle at an ambient temperature of 22±2°C and a relative humidity of 55±10%. All animal IDs are listed in Supp. Table 5. |
| Wild animals | The study did not involve wild animals. |
| Reporting on sex | Samples were obtained from mice from either sex. Sex does not suppose any confounding factor in this study. We provide information on the sex and age of all animals related to all samples in the updated Supplementary Table 5. |
| Field-collected samples | This study did not involve samples collected from the field. |
| Ethics oversight | All animal protocols applying to all mice except M1 were approved by the Ethics Committee of the board of the Francis Crick Institute and the United Kingdom Home Office under the Animals (Scientific Procedures) Act 1986. For M1, animal experiments were approved by the cantonal veterinary office of the canton Aargau (Switzerland) and were carried out in accordance with the Swiss law on animal protection. All animal IDs are listed in Supp. Table 5. |

Note that full information on the approval of the study protocol must also be provided in the manuscript.

