## [Peer Review File · Nature Methods]

Non-destructive X-ray tomography of brain tissue ultrastructure

Corresponding Author: Professor Andreas Schaefer

Version 0:

Decision Letter:

10th Jan 2025

Dear Andreas,

Thank you for your patience. Your Article, "Non-destructive X-ray tomography of brain tissue ultrastructure", has now been seen by two reviewers. As you will see from their comments below, although the reviewers find your work of considerable potential interest, they have raised a number of concerns. We are interested in the possibility of publishing your paper in Nature Methods, but would like to consider your response to these concerns before we reach a final decision on publication.

We therefore invite you to revise your manuscript to address these concerns. Importantly, we think it will be important to add a demonstration showing that (semi-)automatic segmentation is feasible.

Link Redacted

We hope to receive your revised paper within 2-3 months. If you cannot send it within this time, please let us know. In this event, we will still be happy to reconsider your paper at a later date so long as nothing similar has been accepted for publication at Nature Methods or published elsewhere.

OPEN SCIENCE REQUIREMENTS

REPORTING SUMMARY AND EDITORIAL POLICY CHECKLISTS

EXTENDED DATA FIGURES

DATA AVAILABILITY

All novel DNA and RNA sequencing data, protein sequences, genetic polymorphisms, linked genotype and phenotype data, gene expression data, macromolecular structures, and proteomics data must be deposited in a publicly accessible database, and accession codes and associated hyperlinks must be provided in the "Data Availability" section.

CODE AVAILABILITY

Please include a "Code Availability" subsection in the Online Methods which details how your custom code is made available. Only in rare cases (where code is not central to the main conclusions of the paper) is the statement "available upon request" allowed (and reasons should be specified).

For more information on our code sharing policy and requirements, please see:
<https://www.nature.com/nature-research/editorial-policies/reporting-standards#availability-of-computer-code>

MATERIALS AVAILABILITY

ORCID

Best regards,
Nina

Nina Vogt, PhD
Senior Editor
Nature Methods

Reviewers' Comments:

Reviewer #1 (Remarks to the Author):

Current Issue:

Understanding cellular, tissue, and organ function relies on subcellular mapping typically conducted by techniques which reveal biological ultrastructure. The gold-standard method of ultrastructural assessment is electron microscopy, and for large tissue segments, scanning electron microscopy. However, electrons only penetrate at most a few hundred nanometers, thus samples must be sectioned into smaller sections prior to imaging. This imposes long sample processing times and potential for tissue damage and information loss during typical sample milling processes, such as focused ion beam milling.

Thus, an illumination methodology which could penetrate thick biological samples while still providing ultrastructural information would be greatly beneficial to the assessment of cellular, tissue, and organ structure and function. X-rays represent a potential candidate, as they can be generated at Angstrom-scale wavelengths and can penetrate tissue up to the cm-scale. Unfortunately, traditional X-rays damage biological tissue with prolonged imaging and cannot provide high resolution due to low brilliance, and thus would be unable to provide high resolution images of larger, millimeter scale tissue blocks.

New Findings:

- High-brilliance and highly-resolving 4th generation synchrotron X-ray sources reveal ultrastructure of millimeter-scale brain tissue volumes
- A rotational cryogenic sample stage, high-precision interferometric positioners delivering 4th generation synchrotron X-rays, and a novel X-ray irradiation-resistant epoxy resin facilitate high resolution and non-destructive X-ray ptychography, capable of generating a series of reconstructed 2D phase projections from tissue blocks
- Computational advancements aimed at limiting residual sample deformations enable accurate 3D reconstructed tomograms

of tissue ultrastructure with isotropic 40 nm resolution

Thus, the authors propose a new method of X-ray ptychography which provides 40 nm isotropic resolution of brain tissue blocks, capable of revealing subcellular structure of neurons. The authors can identify and characterize neurites, axons, dendrites, and synapses—subcellular compartments indispensable for neuronal and brain function.

We are enthusiastic about the manuscript, but several concerns are noted.

Concerns:

Major:

1. Usefulness.

- For this approach to be useful in the community, it is imperative to demonstrate that images can be segmented, at least, in a semi-automated fashion. For the connectomics at the ultrastructural level, being able to identify synapses is essential, but more importantly, all the fine neuronal and glial processes will need to be segmented and traced since this is the information that we cannot gain from fluorescence imaging. Thus, as a method paper, such demonstration is needed.

- Although possible, ROTO staining is mostly compatible with tissues prepared by conventional chemical fixation approach. For fine ultrastructural preservation, cryo-preservation is necessary (<https://pubmed.ncbi.nlm.nih.gov/26259873/>). It is unclear how compatible this method is to samples prepared by high-pressure freezing. Authors should at the least discuss compatibility.

2. Extracellular space. The TGPAP-embedded sample shown in Figure 4g and 5i has much more extracellular space than those embedded in other resin. The paper referenced above (<https://pubmed.ncbi.nlm.nih.gov/26259873/>) describes how chemical fixation causes shrinkage of tissues and results in the loss of extracellular space. Can authors explain why more extracellular spaces are present in these tissues? If this resin is capable of preserving the morphology better, people in the field would need to consider using this resin for all ultrastructural studies. Can this resin be sectioned on the diamond knife properly (down to ~40 nm)? Authors do discuss the use of hot knife ultramicrotomy in the Discussion.

3. Manuscript structure:

-Figure 2

The authors introduce the TGPAP resin in Figure 2c, yet do not mention it when discussing the other panels within Figure 2. Thus, this Figure becomes hard for the reader to discern, particularly when trying to understand sample preparation differences. The authors should fully describe the figure when it is introduced in the results section, even if they wish to bury the introduction of the TGPAP resin later in the manuscript.

-Figure 2d

This graph is clunky from a reader's perspective. The authors should show instead separated data in Supp Fig. 2.4

-Figure 5

TGPAP is finally introduced in this figure, and at this point in the manuscript, but it is used and mentioned previously. For manuscript coherence, TGPAP should be introduced prior to its first use in Figure 2. If the desire is to fully introduce it later, a simple mention of the resins used and basic properties (e.g. TGPAP, a radiation resistant resin which enables higher doses of X-ray) prior to figure 2, with elaboration later in the text for figure 5 would suffice. Lastly, Figure 5a is never called out in text.

-In my opinion, the manuscript could be greatly improved in flow by simply moving figure 2 to the end of the manuscript and making it figure 5. Thus, making Figure 3 Figure 2, Figure 4 Figure 3, and Figure 5 Figure 4. Alternatively, Figure 2 can be moved to the supplement. Also, in its place, Supp. Fig 2.1 could be changed to Figure 2, which provides a nice presentation of the nonrigid reconstruction method, which the authors highlight as a major advancement in their abstract.

4. Technical concerns:

-Figure 4

This figure aims to benchmark X-ray ptychography to existing methods of ultrastructural detection. While convincing, this section can be improved. In this figure, a data set is presented using TGPAP, but this dataset is never mentioned in text. Further, it is not clear if the following benchmarking analysis in Figure 4d-f uses the datasets from a, g, or both. It is possible that benchmarking results would improve if the authors used purely TGPAP samples, which boast higher resolution. In fact, a comparison between Embed812 samples and TGPAP samples here would be strong evidence that TGPAP provides significant improvement in sample quality and ease/reliability of ultrastructural annotation. Thus, benchmarking should be repeated separately in the TGPAP datasets in g and the Embed812 datasets in a, and subsequently compared.

Minor:

Annotation, grammar, spelling, typing errors:

-line 168

Authors need to state what the ROTO acronym is, as it is the first time it is used in the manuscript.

Reviewer #2 (Remarks to the Author):

The article proposes an approach that combines X-ray ptychography with the study of the optimal staining protocol to protect biological samples from radiation damage. In particular, using these advances, sub-40 nm isotropic resolution has been achieved, enabling detailed imaging of neural structures and the technique has been validated against focused ion beam scanning electron microscopy.

-The paper does not sufficiently discuss in the INTRODUCTION the limitation of the technique and the comparison with the state of the art. The citations referenced in the text are quite dated (i.e. 1999) and lack more recent developments in a rapidly evolving field. In this context, the sentence in line 33 should be revised and expanded. Moreover verify that the comparison with other techniques involves the sample dimension.

-The dimension of the samples should be written in the results and in the methods sections, please verify.

-The paper will only be able to sufficiently demonstrate the innovativeness of the approach (for example with respect to EM) if it addresses the issue of the very small field of view and the challenge of stitching together images captured at 10 microns by 10 microns with wider images that place the details in a broader context.

-Given the importance of the detail that is being highlighted, I do not believe that validation through FIB-SEM is sufficient. I believe that a comprehensive validation of synapses should encompass both structural and chemical aspects: immunohistochemistry or the use of genetically encoded fluorescent proteins should be included.

Minor

Please insert for convenience the scale bar in figures 4b and 5b.

Version 1:

Decision Letter:

Our ref: NMETH-A58253A

17th Jun 2025

Dear Andreas,

Thank you for submitting your revised manuscript "Non-destructive X-ray tomography of brain tissue ultrastructure" (NMETH-A58253A). It has now been seen by the original referees and their comments are below. The reviewers find that the paper has improved in revision, and therefore we'll be happy in principle to publish it in Nature Methods, pending minor revisions to satisfy the referees' final requests and to comply with our editorial and formatting guidelines.

TRANSPARENT PEER REVIEW

ORCID

Thank you again for your interest in Nature Methods. Please do not hesitate to contact me if you have any questions. We will

be in touch again soon.

Best regards,
Nina

Nina Vogt, PhD
Senior Editor
Nature Methods

Reviewer #1 (Remarks to the Author):

The authors promptly addressed my concerns.

Reviewer #2 (Remarks to the Author):

Dear Editor,

I am writing to provide my response regarding the revised manuscript that I previously reviewed for Nature Methods.

I am pleased to report that the authors have comprehensively addressed all the concerns and questions I raised during the initial review process. The revisions demonstrate a thorough and thoughtful approach to improving the manuscript quality. All the issues I highlighted have been resolved in an excellent manner, and the authors have implemented the suggested improvements effectively. The revised manuscript shows significant enhancement in clarity, methodology, and presentation compared to the original submission.

The paper has been substantially improved and, from my perspective as a reviewer, is now ready for publication. The authors have clearly taken the feedback seriously and have produced a manuscript that meets the high standards expected for Nature Methods.

I recommend acceptance of this revised manuscript for publication.

Thank you for the opportunity to review this work.

Sincerely,

Alessia Cedola

Version 2:

Decision Letter:

1st Oct 2025

Dear Andreas,

I am pleased to inform you that your Article, "Non-destructive X-ray tomography of brain tissue ultrastructure", has now been accepted for publication in Nature Methods. The received and accepted dates will be October 14th, 2024 and October 1st, 2025. This note is intended to let you know what to expect from us over the next month or so, and to let you know where to address any further questions.

Over the next few weeks, your paper will be copyedited to ensure that it conforms to Nature Methods style. Once your paper is typeset, you will receive an email with a link to choose the appropriate publishing options for your paper and our Author Services team will be in touch regarding any additional information that may be required. It is extremely important that you let us know now whether you will be difficult to contact over the next month. If this is the case, we ask that you send us the contact information (email, phone and fax) of someone who will be able to check the proofs and deal with any last-minute problems.

Authors may need to take specific actions to achieve compliance with funder and institutional open access mandates.

If your research is supported by a funder that requires immediate open access (e.g. according to <https://www.springernature.com/gp/open-science/plan-s-compliance>) Plan S principles or the <https://www.springernature.com/gp/open-science/us-federal-agency-compliance> NIH public access policy) then you should select the gold OA route, and we will direct you to the compliant route where possible. Because authors warrant under our subscription licensing terms that they haven't committed to licensing any version of their article under a licence inconsistent with the terms of our agreement – including the applicable embargo period – publication under the subscription model isn't suitable for authors whose funders require no embargo.

If you are active on Twitter/X or Bluesky, please e-mail me your and your coauthors' handles so that we may tag you when the paper is published.

Best regards,
Nina

Nina Vogt, PhD
Senior Editor
Nature Methods

** Visit the Springer Nature Editorial and Publishing website at http://editorial-jobs.springernature.com?utm_source=ejP_NMeth_email&utm_medium=ejP_NMeth_email&utm_campaign=ejp_Nmeth or www.springernature.com/editorial-and-publishing-jobs for more information about our career opportunities. If you have any questions please click [here](mailto:editorial.publishing.jobs@springernature.com).**

Point-by-point reply to the reviewer's comments

Editors' comments:

Thank you for your patience. Your Article, "Non-destructive X-ray tomography of brain tissue ultrastructure", has now been seen by two reviewers. As you will see from their comments below, although the reviewers find your work of considerable potential interest, they have raised a number of concerns. We are interested in the possibility of publishing your paper in Nature Methods, but would like to consider your response to these concerns before we reach a final decision on publication.

We therefore invite you to revise your manuscript to address these concerns. Importantly, we think it will be important to add a demonstration showing that (semi-)automatic segmentation is feasible.

Thank you for your encouragement and interest. We have now conducted a series of new experiments and analyses (including segmentation), addressing all reviewer comments. Please find below a point-by-point response.

Reviewers' Comments:

Reviewer #1 (Remarks to the Author):

Current Issue:

Understanding cellular, tissue, and organ function relies on subcellular mapping typically conducted by techniques which reveal biological ultrastructure. The gold-standard method of ultrastructural assessment is electron microscopy, and for large tissue segments, scanning electron microscopy. However, electrons only penetrate at most a few hundred nanometers, thus samples must be sectioned into smaller sections prior to imaging. This imposes long sample processing times and potential for tissue damage and information loss during typical sample milling processes, such as focused ion beam milling.

Thus, an illumination methodology which could penetrate thick biological samples while still providing ultrastructural information would be greatly beneficial to the assessment of cellular, tissue, and organ structure and function. X-rays represent a potential candidate, as they can be generated at Angstrom-scale wavelengths and can penetrate tissue up to the cm-scale. Unfortunately, traditional X-rays damage biological tissue with prolonged imaging and cannot provide high resolution due to low brilliance, and thus would be unable to provide high resolution images of larger, millimeter scale tissue blocks.

New Findings:

- *High-brilliance and highly-resolving 4th generation synchrotron X-ray sources reveal ultrastructure of millimeter-scale brain tissue volumes*

- *A rotational cryogenic sample stage, high-precision interferometric positioners delivering 4th generation synchrotron X-rays, and a novel X-ray irradiation-resistant epoxy resin facilitate high resolution and non-destructive X-ray ptychography, capable of generating a series of reconstructed 2D phase projections from tissue blocks*

- *Computational advancements aimed at limiting residual sample deformations enable accurate 3D reconstructed tomograms of tissue ultrastructure with isotropic 40 nm resolution*

Thus, the authors propose a new method of X-ray ptychography which provides 40 nm isotropic resolution of brain tissue blocks, capable of revealing subcellular structure of neurons. The authors can identify and characterize neurites, axons, dendrites, and synapses—subcellular compartments indispensable for neuronal and brain function.

We are enthusiastic about the manuscript, but several concerns are noted.

Thank you for your enthusiasm, encouragement, detailed assessment and constructive comments. Below we provide new data and analyses to address those.

Concerns:

Major:

1. Usefulness.

- For this approach to be useful in the community, it is imperative to demonstrate that images can be segmented, at least, in a semi-automated fashion. For the connectomics at the ultrastructural level, being able to identify synapses is essential, but more importantly, all the fine neuronal and glial processes will need to be segmented and traced since this is the information that we cannot gain from fluorescence imaging. Thus, as a method paper, such demonstration is needed.

Segmentation is indeed the end goal of any volume imaging approach, in particular so for connectomics as pointed out by the reviewer. While most of the published connectomics work until recently relied predominantly on manual skeletonisation or (semi-automated) segmentation (Ding, Smith et al. 2016, Gour, Boergens et al. 2021, Witvliet, Mulcahy et al. 2021), the field is moving towards increasingly automatic segmentation in combination with manual curation and proofreading (Schmidt, Motta et al. 2024, Shapson-Coe, Januszewski et al. 2024, Bae, Baptiste et al. 2025). It is important to note - as the reviewer also suggests - that all segmentation efforts to date require extensive manual proof-reading.

Thus, we went on to assess which aspects of the Ptycho-tomography (PXCT) datasets can already be segmented (beyond synapses - see also below for new data on synapse identification - and without difficult-to-quantify manual proofreading). We performed automated dense

segmentation of both the PXCT and the corresponding FIBSEM datasets. Specifically, we employed pre-existing segmentation models without further fine-tuning or proofreading.

The (8-bit) PXCT image intensity range was stretched, so that values of 70 mapped to 0 and 120 mapped to 255. An affinity-based segmentation pipeline (Lee, Zung et al. 2017, Macrina, Lee et al. 2021) was applied using a pre-existing segmentation model that had been trained to produce affinities and dense segmentation (**Fig R_Seg1**) using a mouse cerebellum data collected with FIBSEM, where the input and output resolution were 16 nm isotropic. The model was applied to an isotropic 14 nm downsampled FIBSEM dataset, and to the PXCT data at native 28 nm isotropic voxel size. Various watershed and agglomeration parameters were tested and manually inspected to identify a reasonable best-effort segmentation (**Fig R_seg1**). The resulting best-effort segmentation can be viewed here: <https://spelunker.cave-explorer.org/#!middleauth+https://global.daf-apis.com/nglstate/api/v1/5950833582669824> (a free login needs to be created to view the data).

Overall, already the out-of-the-box EM-derived segmentation algorithms could fully automatically segment not only the FIBSEM data but also provided segmentation for the PXCT datasets. Thicker neurites (**Fig R_Seg2**) or higher contrast features such as mitochondria (**Fig R_Seg3**) could be segmented quite accurately. Thin neurites suffered from false splits and mergers relative to the FIBSEM dataset (**Fig R_Seg_4,5**).

It is encouraging that EM-derived automatic segmentation can be applied to the PXCT datasets. In order to reliably recover neurites with sufficiently small rates of false mergers and splits, further improvements will be needed for PXCT acquisition and/or processing. The ability to perform correlated FIBSEM experiments will in turn simplify the analysis by providing ground truth (although of course also FIB-SEM segmentations still require significant manual proofreading).

Figure R_Seg1: Dense proposal segmentation of the FIBSEM dataset (left) and the PXCT dataset (right) generated by the same pre-existing FIBSEM-derived segmentation model without fine-tuning or proofreading.

Figure R_Seg2: Thick neurites tend to be segmented well in both the PXCT dataset (right) as well as the FIBSEM dataset (left). Note, however, examples of residual mergers and splits in the PXCT dataset.

Figure R_Seg3: Mitochondria segmentation.

Figure R_Seg4 : Thin branches can be missing (top) or split (bottom) in the PXCT dataset (right) compared to the FIBSEM dataset (left).

Figure R_Seg5: Thin neurites can cause false mergers between neighboring non-connected objects in the PXCT dataset (right) compared to the FIBSEM dataset (left).

We now have included a new figure Supp Fig. 6.1 on segmentation and provide the dense segmentations to the readers in a fully open and browsable way. We also discuss the challenges and developments needed for reliable automatic segmentation in the revised manuscript (lines 286ff).

- Although possible, ROTO staining is mostly compatible with tissues prepared by conventional chemical fixation approach. For fine ultrastructural preservation, cryo-preservation is necessary (<https://pubmed.ncbi.nlm.nih.gov/26259873/>). It is unclear how compatible this method is to samples prepared by high-pressure freezing. Authors should at the least discuss compatibility.

The reviewer makes an important point here that for many applications, in particular outside the connectomics field, ultrastructure preservation without chemical fixation is desirable. For the analysis of neurites, synapses and general connectivity structure of brain tissue, chemical fixation followed by heavy-metal staining has been the method of choice and is thought to leave ultrastructural features sufficiently intact for accurate reconstruction of circuitry.

Ptycho-tomography is indeed compatible with cryopreserved samples, including those prepared with high-pressure freezing (Guizar-Sicairos, Diaz et al. 2011) or frozen-hydrated samples (Shahmoradian, Tsai et al. 2017, Tran, Tsai et al. 2020). Our approach, combining ptycho-tomography at cryo-temperature with non-rigid reconstructions and highly X-ray radiation resistant embedding resins to reach sub 40 nm resolution, should also be applicable to any soft tissue sample that is heavy metal-stained and resin-embedded (provided that they are embedded in the resin we describe, TGPAP-DDM).

We have not explicitly explored alternative infiltration options for TGPAP-DDM beyond the described one, which requires heavy metal-stained, dehydrated samples to be at room temperature for the resin infiltration to start. Nevertheless, we think that the current protocol should already be compatible with high-pressure-frozen samples.

High-pressure freezing (HPF) is a method to fix biological soft tissue samples of up to 100-200 μ m in thickness across their shortest dimension. This method provides a fast freezing process that generates vitreous ice instead of crystalline ice forms (that would otherwise break ultrastructure). The kinetics of temperature diffusion through the sample at the high pressures achievable define the deepest region in the sample where vitreous ice can be achieved, which is typically thought to be 50-100 μ m. This dimension makes HPF a convenient method for the study of living specimens of <100 μ m in size - cells, microorganisms and developmental stages of small organisms (like embryos or larvae).

HPF-frozen cells and tissues are brought from the cryoprotectant medium at -90°C to a carrier medium at 0°C, through a process known as freeze-substitution. At that point, heavy metal staining can take place.

After being stained with heavy metals, tissues can be infiltrated with epoxy resins at 0°C (for resins that, unpolymerised, are liquid enough at that temperature) or alternatively warmed up to room temperature and infiltrated with standard resins (such as the mixture EMBED-812, Araldite 502 and DDSA; Liu, Pokrovskaya et al. 2023).

Since we have not characterised the viscosity of TGPAP-DDM at 0°C, we would recommend bringing the samples to room temperature before embedding.

Note that this approach at this point stems from HPF-fixed samples that have been metal-stained with osmium before infiltration. Osmium is a strong fixative, and could be a critical agent preserving ultrastructure in the warm-up process to room temperature. While it might be possible to preserve ultrastructure in HPF-fixed, not metal-stained samples embedded at room temperature, this protocol remains to be developed. We foresee a critical point in minimising degradation of ultrastructure in the not metal-stained sample at non-cryo temperatures. This minimisation could involve ensuring this occurs in a short time window (infiltration following immediately after sample thawing, not storing non-infiltrated samples at room temperature).

We now discuss extensions of this work to samples without chemical fixation in lines 347ff of the revised manuscript.

2. Extracellular space. The TGPAP-embedded sample shown in Figure 4g and 5i has much more extracellular space than those embedded in other resin. The paper referenced above (<https://pubmed.ncbi.nlm.nih.gov/26259873/>) describes how chemical fixation causes shrinkage of tissues and results in the loss of extracellular space. Can authors explain why more extracellular spaces are present in these tissues?

Extracellular space (ECS) is indeed a key parameter for both staining of samples, as well as ease of segmentation (Pallotto, Watkins et al. 2015). Moreover, traditional fixation protocols have sometimes resulted in severely reduced ECS compared to the *in vivo* state as the reviewer points out.

To address these concerns, we have now quantified ECS in our sample (see **Fig. R_ECS_1**) and provide a comparison to published protocols.

Figure R_ECS_1: Quantification of ECS by manual segmentation of $n=3$ ROIs, each covering a volume of $3000 \times 3000 \times 80 \text{ nm}^3$ ($375 \times 375 \times 10 \text{ voxels}^3$ at $(8 \text{ nm})^3$) resulting in an estimate of $16 \pm 2\%$ ECS.

Our sample was dissected in an isoosmotic buffer (300+-20 mOsm/kg), immersion-fixed in fixative diluted in a vehicle buffer at the same osmolarity and kept at that osmolarity until staining with Osmium.

ECS is affected by osmotic pressure (e.g. Van Harreveld and Khattab 1967, Pallotto, Watkins et al. 2015, Lu, Wu et al. 2023): hypertonic buffers lead to shrinkage of cells and hypotonic buffers (like distilled water) lead to swelling of cells in tissues, eventually triggering lysis: washing cells and tissues with distilled water is an established lysis protocol used in molecular biology to extract intracellular proteins and nucleic acids (Boone, Ford et al. 1969, de Weerd, Ruiter et al. 2024).

ECS fraction is also affected by the method used for fixation (Korogod, Petersen et al. 2015, Pallotto, Watkins et al. 2015, see **Fig. R_ECS_2**): perfusion fixation typically depletes ECS.

However, the fixation method is likely to modulate ECS also via osmolarity although through an indirect manner: with the blood brain barrier acting as a semipermeable membrane, intracardiac perfusion of fixative with osmolarity matched using an impermeable solvent can lead to osmotic mismatch through the blood brain barrier, eventually delivering a hypotonic buffer that triggers absorption of the ECS by cells, thereby reducing the preserved ECS fraction.

It has been recently shown that a multi-step intracardiac perfusion protocol, delivering a sucrose gradient prior to fixative perfusion, can permeabilise sufficiently the blood brain barrier to the extent of preserving extracellular space in mouse brain tissue (Lu, Wu et al. 2023).

Osmolarity matching delivers extracellular space preservation in immersion-fixed samples, by delivering fixatives in a vehicle buffer that is isoosmotic to the organism. The osmolarity changes across tissues and species: plants, fish and human tissues will have different osmolarities at resting state than the rodent brain, which has an osmolarity of 300 mOsm/kg. Therefore, protocols aiming to preserve the extracellular space of soft tissues in plant, fish and human samples will need to match a different osmolarity. Protocols matching the osmolarity of the mouse brain will therefore provide hypertonic or hypotonic media to those samples - leading to excessive shrinkage or lysis.

Our samples of mouse brain external plexiform layer display a $16 \pm 2\%$ of extracellular space (**Fig. R_ECS_1**). Directly comparing this with the published literature analysing ECS for different fixation protocols for mouse brain (**Fig R_ECS_2** and Korogod, Petersen et al. 2015, Pallotto, Watkins et al. 2015) shows that these are quite consistent with similarly prepared samples (immersion-fixed with an isosmotic buffer).

We now include the quantification of ECS as a new supplementary figure **Supp. Fig. 5.4** in the paper and on lines 285ff.

If this resin is capable of preserving the morphology better, people in the field would need to consider using this resin for all ultrastructural studies. Can this resin be sectioned on the diamond knife properly (down to ~40 nm)? Authors do discuss the use of hot knife ultramicrotomy in the Discussion.

TGPAP indeed preserves ultrastructural well as confirmed by our FIBSEM studies. We would not want to claim any superiority in ultrastructure preservation to other resins commonly used in volume EM or general EM studies (Durcupan, Epon, EMBED812 etc). Regarding cutting performance, one of the key reasons for us to invest into developing X-ray protocols for nano-resolution volume tomography is that this will remove the pressure on cutting large series of ultrathin sections. Thus, we have not aimed to develop a resin compatible for both serial-section (or serial-blockface) volume EM *and* X-ray tomography. Nevertheless, we were happy to find that FIBSEM analysis is possible with minimal artefacts for TGPAP embedded samples.

We acknowledge that cutting sections can nevertheless be useful for assessing ultrastructure in a rapid manner. We therefore collected 60-80 nm sections with a diamond knife for SEM analysis (to assess resin penetration prior to in depth PXCT analysis). While TGPAP is not necessarily optimal for reliable serial sectioning at 50 nm or below, we were able to acquire sections with sufficient quality for assessing ultrastructural integrity (**Fig. R_Section**).

Unlike other epoxy resins such as EMBED812 that are commonly used in EM, our TGPAP formulation has specifically been optimized for radiation hardness, but not for ultrathin serial sectioning. Routine, artifact-free cutting and collection of TGPAP serial sections thinner than 100

nm will likely require optimization of the TGPAP formulation - which could impact the radiation hardness.

We now discuss sectioning in the Discussion of the revised manuscript (lines 353ff.)

Figure R_section: Example SEM images of 60-80 nm thin single sections cut with a Leica UC7 ultramicrotome from the same TGPAP sample block as the pillars shown in Figure 4h and 4j of the original manuscript (now Figure 5g and 5j in the revised manuscript). Top: High magnification image of the central part in the low magnification overview image (bottom).

3. Manuscript structure:

-Figure 2

The authors introduce the TGPAP resin in Figure 2c, yet do not mention it when discussing the other panels within Figure 2. Thus, this Figure becomes hard for the reader to discern, particularly when trying to understand sample preparation differences. The authors should fully describe the figure when it is introduced in the results section, even if they wish to bury the introduction of the TGPAP resin later in the manuscript.

We very much appreciate these suggestions. We must admit that - when re-reading the manuscript - we very much agree. We have therefore switched figures around, first introducing TGPAP. We think that the paper has become substantially clearer with that.

-Figure 2d

This graph is clunky from a reader's perspective. The authors should show instead separated data in Supp Fig. 2.4

Thank you for pointing out this presentation issue. We have followed the reviewer's advice and present a simplified diagram in the figure: We have added graphs split by resin type and simplified the individual panels in the main figure (new **Figure 3**). We are convinced that this has simplified and improved the figure.

-Figure 5

TGPAP is finally introduced in this figure, and at this point in the manuscript, but it is used and mentioned previously. For manuscript coherence, TGPAP should be introduced prior to its first use in Figure 2. If the desire is to fully introduce it later, a simple mention of the resins used and basic properties (e.g. TGPAP, a radiation resistant resin which enables higher doses of X-ray) prior to figure 2, with elaboration later in the text for figure 5 would suffice. Lastly, Figure 5a is never called out in text.

-In my opinion, the manuscript could be greatly improved in flow by simply moving figure 2 to the end of the manuscript and making it figure 5. Thus, making Figure 3 Figure 2, Figure 4 Figure 3, and Figure 5 Figure 4. Alternatively, Figure 2 can be moved to the supplement. Also, in its place, Supp. Fig 2.1 could be changed to Figure 2, which provides a nice presentation of the nonrigid reconstruction method, which the authors highlight as a major advancement in their abstract.

Thank you again - as mentioned above we have switched the order around, essentially following the reviewer's suggestion, resulting in a - as we think - much improved paper. We decided to leave **Supp. Fig. 2.1** (originally **Supp. Fig. 5.1**) as supplement to allow for the detailed description that we think useful but have moved the description of the resin to **Figure 2** (originally **Fig. 5**), followed by an analysis of the dose-dependence of resolution across the entire range of doses. We then go on to assess the identification of biological features (including an expanded figure about synapse identification in both dose-regimes, see below) before ending on a more detailed analysis of ultrastructure at the highest dose / resolution.

Thank you again for these very helpful suggestions.

4. Technical concerns:

-Figure 4

This figure aims to benchmark X-ray ptychography to existing methods of ultrastructural detection. While convincing, this section can be improved. In this figure, a data set is presented using TGPAP, but this dataset is never mentioned in text. Further, it is not clear if the following benchmarking analysis in Figure 4d-f uses the datasets from a, g, or both. It is possible that benchmarking results would improve if the authors used purely TGPAP samples, which boast

higher resolution. In fact, a comparison between Embed812 samples and TGPAP samples here would be strong evidence that TGPAP provides significant improvement in sample quality and ease/reliability of ultrastructural annotation. Thus, benchmarking should be repeated separately in the TGPAP datasets in g and the Embed812 datasets in a, and subsequently compared.

The reviewer makes an important point about quantification of improvements in sample preparation and data acquisition - as well as about how specific resolution improvements impact different benchmarks.

We have largely aimed to use the standard resolution measurement of Fourier Shell Correlation to quantify the impact of dose and reconstruction algorithms. We do recognise that while this is a good proxy, ultimately reliable detection of features is key.

As photon flux was limiting at the 3rd generation storage ring of SLS (before the recent upgrade), we reduced sample size for the highest dose. Thus the TGPAP sample presented was a cylinder of 10 μm diameter imaged 5 μm tall (compared to 20 for the Epon embedded one, imaged 20 μm tall). We therefore originally didn't attempt to quantify synapse detectability in the same way.

We have now performed the same analysis for the smaller sample as well. While the smaller sample numbers make the data more noisy, we find that the TGPAP samples tended to result in higher average X-ray scores for synapses detected in EM (**Figure R_SynapseAnnotation**).

We now provide these new data in the new **Figure 5** (formerly **Figure 4**) of the revised manuscript (lines 277ff).

Figure R_SynapseAnnotation. (a,b,c,d): Data presented in the original manuscript for Epon-embedded tissue at an Xray dose of 3.8×10^8 Gy. (e,f,g,h): New annotation results for TGPAP embedded tissue at an X-ray dose of 2.5×10^9 Gy.

Minor:

Annotation, grammar, spelling, typing errors:

-line 168

Authors need to state what the ROTO acronym is, as it is the first time it is used in the manuscript.

Thank you for pointing out this omission. We have now spelled out the abbreviation at first occurrence (lines 118f). We have also re-checked grammar and spelling throughout the text.

Reviewer #2 (Remarks to the Author):

The article proposes an approach that combines X-ray ptychography with the study of the optimal staining protocol to protect biological samples from radiation damage. In particular, using these advances, sub-40 nm isotropic resolution has been achieved, enabling detailed imaging of neural

structures and the technique has been validated against focused ion beam scanning electron microscopy.

We appreciate your encouragement and constructive comments that we address in detail below.

-The paper does not sufficiently discuss in the INTRODUCTION the limitation of the technique and the comparison with the state of the art. The citations referenced in the text are quite dated (i.e. 1999) and lack more recent developments in a rapidly evolving field. In this context, the sentence in line 33 should be revised and expanded. Moreover verify that the comparison with other techniques involves the sample dimension.

Thank you for pointing this out. We have now augmented the introduction accordingly, citing more current X-ray tomography references (e.g. Miao et al 2025, Zhang et al 2024, Azevedo et al 2024, Palermo et al 2025, Kjer et al 2025, Mizutani et al 2023) as well as some recent connectomics references (Bae et al 2025, Pospisil et al 2024) and have expanded the relevant sections. We have also tried to reduce the number of older references where possible (e.g. removed Momose et al 1996). We do, however, where possible try to keep the reference to the original papers demonstrating a new method like ptychography or holotomography, so have opted to specifically keep a reference to the original paper. We have revised the respective sentence and made clear where we refer to the original method compared to the more recent applications for imaging neural tissue.

-The dimension of the samples should be written in the results and in the methods sections, please verify.

This is indeed an important point and we have now included sample dimensions more explicitly, by adding a specific table (**Supp. Table 5**, see below in **TableR_samples**) and in the text (e.g. on lines 163, 203 and 775f. of the revised manuscript).

sampleID	resinID	epoxyID	diameter_um
CX-1	B	Epon812	20.8
CX-2	B	Epon812	21.4
EPL-1	A	EMbed812	22.5
EPL-2	A	EMbed812	22.2
EPL-3	A	EMbed812	22.8
EPL-5	A	EMbed812	22.1
EPL-6	A	EMbed812	10.5
EPL-8	A	EMbed812	23.6

GLOM-F	A	EMbed812	20
RAD-3	A	EMbed812	20.4
M1_1	C	EMbed812	10.76
M1_2	C	EMbed812	10.93
v332_2dot	D	TPTE	10.44
v357_1dot	E	TGPAP	10.68
v357_30mu	E	TGPAP	30
v357_3dot	E	TGPAP	10.52
v359_1dot	C	EMbed812	10.25

TableR_samples. List of samples, now provided as Supp. Table 5.

-The paper will only be able to sufficiently demonstrate the innovativeness of the approach (for example with respect to EM) if it addresses the issue of the very small field of view and the challenge of stitching together images captured at 10 microns by 10 microns with wider images that place the details in a broader context.

Scaling up the approach we have presented here to larger volumes is an important next challenge. We have now collected pilot data to assess how this could be obtained.

We chose to perform additional ptychography measurements on TGPAP embedded brain tissue - but in the laminography geometry. Here, the rotation axis is tilted with respect to the X-ray beam (at 61 degrees rather than orthogonal as in tomography). This makes it possible to analyse samples extending to essentially arbitrary dimensions in X and Y with the thickness in Z determined by X-ray absorption, and depth of field (DOF, see however Li, Wojcik et al. 2017, Jacobsen 2018, Ali, Du et al. 2020, Du, Nashed et al. 2020 for “multi-slice-ptychography” approaches to extend thickness to several multiples of the DOF).

Figure R_Laminography shows our preliminary data. We can indeed reconstruct brain tissue volumes with comparable resolution to tomography. Importantly, synapses are readily visible. We now discuss laminography and the possibility to thereby extend volumes to (essentially) arbitrary lateral dimensions on lines 388ff. of the revised manuscript.

-Given the importance of the detail that is being highlighted, I do not believe that validation through FIB-SEM is sufficient. I believe that a comprehensive validation of synapses should encompass both structural and chemical aspects: immunohistochemistry or the use of genetically encoded fluorescent proteins should be included.

It is indeed a highly active field of research to determine chemical aspects of synapses from ultrastructural data. Here, we largely rely on the experience of the volume EM community.

The mammalian volume EM community has approached this from several angles. One is to “only” measure structural information at high resolution but use prior knowledge about cell types to identify e.g. putative excitatory or inhibitory neurons (e.g. Helmstaedter, Briggman et al. 2013,

Kasthuri, Hayworth et al. 2015, Schmidt, Gour et al. 2017, Gour, Boergens et al. 2021, Witvliet, Mulcahy et al. 2021, Shapson-Coe, Januszewski et al. 2024). This includes estimating the chemical nature of synapses by comparing morphological features in the volumeEM dataset to other, independent studies. There, e.g. synaptic transmission is measured electrophysiologically in combination with morphological analysis of individual neurons or where immunohistochemistry is combined with staining of individual neurons etc. Generally, e.g. the morphology of glutamatergic excitatory neurons or GABAergic interneurons is sufficiently well established to allow reliable circuit identification with “only” structural information.

A second general approach has been to combine structural analysis with functional analysis (e.g. (Briggman, Helmstaedter et al. 2011, Kuan, Bondanelli et al. 2024, Bae, Baptiste et al. 2025, Ding, Fahey et al. 2025, Elabbady, Seshamani et al. 2025, Gamlin, Schneider-Mizell et al. 2025, Schneider-Mizell, Bodor et al. 2025) - e.g. by performing *in vivo* functional analysis prior to structural analysis which has the potential to reveal further insight into the differential function of e.g. synapses (Performing prior *in vivo* functional measurements is possible with X-ray tomography as well as we show in a recent preprint (Zhang, Bosch et al. 2025)).

High-resolution structural analysis, however, was recently shown to directly allow reliable identification of chemical identity of synapses in *Drosophila* (**Fig R_ChemFromStructure** from (Eckstein, Bates et al. 2024)). We expect that systematic studies like this will be performed by the mammalian volumeEM community as well.

In order for X-ray tomography data to reliably predict chemical nature of synapses the strategies will be exactly the same. In fact, as image analysis methods established for volumeEM can readily be used with ptychography data (as essentially the same features are visualised in both approaches), we expect that no substantial adjustment will be needed.

Our aim of this manuscript is to establish how ultrastructural features can be visualised with X-ray tomography (without the need for ultrathin sectioning) - and then to leverage the approaches established by the EM community,

We now discuss these important points in our revised manuscript on lines 377ff.

Minor

Please insert for convenience the scale bar in figures 4b and 5b.

We have now included scale bars in these panels (new **Figures 5b** and **2b**, respectively) and apologise for the earlier omission.

References:

- Ali, S., M. Du, M. F. Adams, B. Smith and C. Jacobsen (2020). "Comparison of distributed memory algorithms for X-ray wave propagation in inhomogeneous media." *Opt Express* **28**(20): 29590-29618.
- Azevedo, A., E. Lesser, J. S. Phelps, B. Mark, L. Elabbady, S. Kuroda, A. Sustar, A. Moussa, A. Khandelwal, C. J. Dallmann, S. Agrawal, S. J. Lee, B. Pratt, A. Cook, K. Skutt-Kakaria, S. Gerhard, R. Lu, N. Kemnitz, K. Lee, A. Halageri, M. Castro, D. Ih, J. Gager, M. Tammam, S. Dorkenwald, F. Collman, C. Schneider-Mizell, D. Brittain, C. S. Jordan, M. Dickinson, A. Pacureanu, H. S. Seung, T. Macrina, W. A. Lee and J. C. Tuthill (2024). "Connectomic reconstruction of a female *Drosophila* ventral nerve cord." *Nature* **631**(8020): 360-368.
- Bae, J. A., M. Baptiste, M. R. Baptiste, C. A. Bishop, A. L. Bodor, D. Brittain, V. Brooks, J. Buchanan, D. J. Bumbarger, M. A. Castro, B. Celii, E. Cobos, F. Collman, N. M. Da Costa, B. Danskin, S. Dorkenwald, L. Elabbady, P. G. Fahey, T. Fliss, E. Froudarakis, J. Gager, C. Gamlin, W. Gray-Roncal, A. Halageri, J. Hebditch, Z. Jia, E. Joyce, J. Ellis-Joyce, C. Jordan, D. Kapner, N. Kemnitz, S. Kinn, L. M. Kitchell, S. Koolman, K. Kuehner, K. Lee, K. Li, R. Lu, T. Macrina, G. Mahalingam, J. Matelsky, S. McReynolds, E. Miranda, E. Mitchell, S. S. Mondal, M. Moore, S. Mu, T. Muhammad, B. Nehoran, E. Neace, O. Ogedengbe, C. Papadopoulos, S. Papadopoulos, S. Patel, G. J. Y. P. Vega, X. Pitkow, S. Popovych, A. Ramos, R. C. Reid, J. Reimer, P. K. Rivlin, V. Rose, Z. M. Sauter, C. M. Schneider-Mizell, H. S. Seung, B. Silverman, W. Silversmith, A. Sterling, F. H. Sinz, C. L. Smith, R. Swannstrom, S. Suckow, M. Takeno, Z. H. Tan, A. S. Tolias, R. Torres, N. L. Turner, E. Y. Walker, T. Wang, A. Wanner, B. A. Wester, G. Williams, S. Williams, K. Willie, R. Willie, W. Wong, J. Wu, C. Xu, R. Yang, D. Yatsenko, F. Ye, W. Yin, R. Young, S.-C. Yu, D. Xenos and C. Zhang (2025). "Functional connectomics spanning multiple areas of mouse visual cortex." *Nature* **640**(8058): 435-447.
- Boone, C. W., L. E. Ford, H. E. Bond, D. C. Stuart and D. Lorenz (1969). "Isolation of plasma membrane fragments from HeLa cells." *J Cell Biol* **41**(2): 378-392.
- Briggman, K. L., M. Helmstaedter and W. Denk (2011). "Wiring specificity in the direction-selectivity circuit of the retina." *Nature* **471**(7337): 183-188.
- de Weerd, S., E. A. Ruitter, E. Calicchia, G. Portale, J. J. Schuringa, W. H. Roos and A. Salvati (2024). "Optimization of Cell Membrane Purification for the Preparation and Characterization of Cell Membrane Liposomes." *Small Methods* **8**(12): e2400498.
- Ding, H., R. G. Smith, A. Poleg-Polsky, J. S. Diamond and K. L. Briggman (2016). "Species-specific wiring for direction selectivity in the mammalian retina." *Nature* **535**(7610): 105-110.
- Ding, Z., P. G. Fahey, S. Papadopoulos, E. Y. Wang, B. Celii, C. Papadopoulos, A. Chang, A. B. Kunin, D. Tran, J. Fu, Z. Ding, S. Patel, L. Ntanavara, R. Froebe, K. Ponder, T. Muhammad, J. A. Bae, A. L. Bodor, D. Brittain, J. Buchanan, D. J. Bumbarger, M. A. Castro, E. Cobos, S. Dorkenwald, L. Elabbady, A. Halageri, Z. Jia, C. Jordan, D. Kapner, N. Kemnitz, S. Kinn, K. Lee, K. Li, R. Lu, T. Macrina, G. Mahalingam, E. Mitchell, S. S. Mondal, S. Mu, B. Nehoran, S. Popovych, C. M. Schneider-Mizell, W. Silversmith, M. Takeno, R. Torres, N. L. Turner, W. Wong, J. Wu, W. Yin, S.-C. Yu, D. Yatsenko, E. Froudarakis, F. Sinz, K. Josić, R. Rosenbaum, H. S. Seung, F. Collman, N. M. Da Costa, R. C. Reid, E. Y. Walker, X. Pitkow, J. Reimer and A. S. Tolias (2025). "Functional connectomics reveals general wiring rule in mouse visual cortex." *Nature* **640**(8058): 459-469.
- Du, M., Y. S. G. Nashed, S. Kandel, D. Gursoy and C. Jacobsen (2020). "Three dimensions, two microscopes, one code: Automatic differentiation for x-ray nanotomography beyond the depth of focus limit." *Sci Adv* **6**(13): eaay3700.
- Eckstein, N., A. S. Bates, A. Champion, M. Du, Y. Yin, P. Schlegel, A. K.-Y. Lu, T. Rymer, S. Finley-May, T. Paterson, R. Parekh, S. Dorkenwald, A. Matsliah, S.-C. Yu, C. Mckellar, A. Sterling, K. Eichler, M. Costa, S. Seung, M. Murthy, V. Hartenstein, G. S. X. E. Jefferis and J. Funke (2024). "Neurotransmitter classification from electron microscopy images at synaptic sites in *Drosophila melanogaster*." *Cell* **187**(10): 2574-2594.e2523.
- Elabbady, L., S. Seshamani, S. Mu, G. Mahalingam, C. M. Schneider-Mizell, A. L. Bodor, J. A. Bae, D. Brittain, J. Buchanan, D. J. Bumbarger, M. A. Castro, S. Dorkenwald, A. Halageri, Z. Jia, C. Jordan, D. Kapner, N. Kemnitz, S. Kinn, K. Lee, K. Li, R. Lu, T. Macrina, E. Mitchell, S. S. Mondal, B. Nehoran, S. Popovych, W. Silversmith, M. Takeno, R. Torres, N. L. Turner, W. Wong, J. Wu, W. Yin, S.-C. Yu, H. S. Seung, R. C. Reid, N. M. Da Costa and F. Collman (2025). "Perisomatic ultrastructure efficiently classifies cells in mouse cortex." *Nature* **640**(8058): 478-486.
- Gamlin, C. R., C. M. Schneider-Mizell, M. Mallory, L. Elabbady, N. Gouwens, G. Williams, A. Mukora, R. Dalley, A. L. Bodor, D. Brittain, J. Buchanan, D. J. Bumbarger, E. Joyce, D. Kapner, S. Kinn, G. Mahalingam, S. Seshamani, M. Takeno, R. Torres, W. Yin, P. R. Nicovich, J. A. Bae, M. A. Castro, S. Dorkenwald, A. Halageri, Z. Jia, C. Jordan, N. Kemnitz, K. Lee, K. Li, R. Lu, T. Macrina, E. Mitchell, S. S. Mondal, S. Mu, B. Nehoran, S. Popovych, W. Silversmith, N. L. Turner, W. Wong, J. Wu, S.-C. Yu, J. Berg, T. Jarsky, B. Lee, H. S. Seung, H. Zeng, R. C. Reid, F. Collman, N. M. Da Costa and S. A. Sorensen (2025). "Connectomics of predicted Sst transcriptomic types in mouse visual cortex." *Nature* **640**(8058): 497-505.
- Gour, A., K. M. Boergens, N. Heike, Y. Hua, P. Laserstein, K. Song and M. Helmstaedter (2021). "Postnatal connectomic development of inhibition in mouse barrel cortex." *Science* **371**(6528).
- Guizar-Sicairos, M., A. Diaz, M. Holler, M. S. Lucas, A. Menzel, R. A. Wepf and O. Bunk (2011). "Phase tomography from x-ray coherent diffractive imaging projections." *Opt Express* **19**(22): 21345-21357.

Helmstaedter, M., K. L. Briggman, S. C. Turaga, V. Jain, H. S. Seung and W. Denk (2013). "Connectomic reconstruction of the inner plexiform layer in the mouse retina." *Nature* **500**(7461): 168-174.

Holler, M., M. Odstrcil, M. Guizar-Sicairos, M. Lebugle, E. Müller, S. Finizio, G. Tinti, C. David, J. Zusman, W. Unglaub, O. Bunk, J. Raabe, A. F. J. Levi and G. Aeppli (2019). "Three-dimensional imaging of integrated circuits with macro-to nanoscale zoom." *Nature Electronics* **2**(10): 464-470.

Jacobsen, C. (2018). "Relaxation of the Crowther criterion in multislice tomography." *Optics Letters* **43**(19): 4811.

Januszewski, M. and V. Jain (2024). "Next-generation AI for connectomics." *Nat Methods* **21**(8): 1398-1399.

Kasthuri, N., K. J. Hayworth, D. R. Berger, R. L. Schalek, J. A. Conchello, S. Knowles-Barley, D. Lee, A. Vazquez-Reina, V. Kaynig, T. R. Jones, M. Roberts, J. L. Morgan, J. C. Tapia, H. S. Seung, W. G. Roncal, J. T. Vogelstein, R. Burns, D. L. Sussman, C. E. Priebe, H. Pfister and J. W. Lichtman (2015). "Saturated Reconstruction of a Volume of Neocortex." *Cell* **162**(3): 648-661.

Kjer, H. M., M. Andersson, Y. He, A. Pacureanu, A. Daducci, M. Pizzolato, T. Salditt, A. L. Robisch, M. Eckermann, M. Topperwien, A. Bjorholm Dahl, M. L. Elkjaer, Z. Illes, M. Pfitz, V. Andersen Dahl and T. B. Dyrby (2025). "Bridging the 3D geometrical organisation of white matter pathways across anatomical length scales and species." *Elife* **13**.

Miao, J. (2025). "Computational microscopy with coherent diffractive imaging and ptychography." *Nature* **637**(8045): 281-295.

Korogod, N., C. C. Petersen and G. W. Knott (2015). "Ultrastructural analysis of adult mouse neocortex comparing aldehyde perfusion with cryo fixation." *Elife* **4**.

Kuan, A. T., G. Bondanelli, L. N. Driscoll, J. Han, M. Kim, D. G. C. Hildebrand, B. J. Graham, D. E. Wilson, L. A. Thomas, S. Panzeri, C. D. Harvey and W.-C. A. Lee (2024). "Synaptic wiring motifs in posterior parietal cortex support decision-making." *Nature*.

Lee, K., J. Zung, P. Li, V. Jain and H. S. Seung (2017). "Superhuman Accuracy on the SNEMI3D Connectomics Challenge." *arXiv*.

Li, K., M. Wojcik and C. Jacobsen (2017). "Multislice does it all-calculating the performance of nanofocusing X-ray optics." *Opt Express* **25**(3): 1831-1846.

Liu, S., I. D. Pokrovskaya and B. Storrie (2023). High-Pressure Freezing Followed by Freeze Substitution: An Optimal Electron Microscope Technique to Study Golgi Apparatus Organization and Membrane Trafficking. *Methods in Molecular Biology*, Springer US: 211-223.

Lu, X., Y. Wu, R. L. Schalek, Y. Meirovitch, D. R. Berger and J. W. Lichtman (2023). "A Scalable Staining Strategy for Whole-Brain Connectomics." *bioRxiv*.

Macrina, T., K. Lee, R. Lu, N. L. Turner, J. Wu, S. Popovych, W. Silversmith, N. Kemnitz, J. A. Bae, M. A. Castro, S. Dorkenwald, A. Halageri, Z. Jia, C. Jordan, K. Li, E. Mitchell, S. S. Mondal, S. Mu, B. Nehoran, W. Wong, S.-c. Yu, A. L. Bodor, D. Brittain, J. Buchanan, D. J. Bumbarger, E. Cobos, F. Collman, L. Elabbady, P. G. Fahey, E. Froudarakis, D. Kapner, S. Kinn, G. Mahalingam, S. Papadopoulos, S. Patel, C. M. Schneider-Mizell, F. H. Sinz, M. Takeno, R. Torres, W. Yin, X. Pitkow, J. Reimer, A. S. Tolias, R. C. Reid, N. M. d. Costa and H. S. Seung (2021). "Petascale neural circuit reconstruction: automated methods." *bioRxiv*.

Mizutani, R., R. Saiga, Y. Yamamoto, M. Uesugi, A. Takeuchi, K. Uesugi, Y. Terada, Y. Suzuki, V. De Andrade, F. De Carlo, S. Takekoshi, C. Inomoto, N. Nakamura, Y. Torii, I. Kushima, S. Iritani, N. Ozaki, K. Oshima, M. Itokawa and M. Arai (2023). "Structural aging of human neurons is opposite of the changes in schizophrenia." *PLoS One* **18**(6): e0287646.

Momose, A., T. Takeda, Y. Itai and K. Hirano (1996). "Phase-contrast X-ray computed tomography for observing biological soft tissues." *Nat Med* **2**(4): 473-475.

Palermo, F., N. Marrocco, L. Dacomo, E. Grisafi, V. Moresi, A. Sanna, L. Massimi, M. Musella, L. Maugeri, I. Bukreeva, F. Fiordaliso, A. Corbelli, O. Junemann, M. Eckermann, P. Cloetens, T. Weitkamp, G. Gigli, N. K. de Rosbo, C. Balducci and A. Cedola (2025). "Investigating gut alterations in Alzheimer's disease: In-depth analysis with micro- and nano-3D X-ray phase contrast tomography." *Sci Adv* **11**(5): eadr8511.

Pallotto, M., P. V. Watkins, B. Fubara, J. H. Singer and K. L. Briggman (2015). "Extracellular space preservation aids the connectomic analysis of neural circuits." *Elife* **4**: e08206.

Pospisil, D. A., M. J. Aragon, S. Dorkenwald, A. Matsliah, A. R. Sterling, P. Schlegel, S. C. Yu, C. E. McKellar, M. Costa, K. Eichler, G. Jefferis, M. Murthy and J. W. Pillow (2024). "The fly connectome reveals a path to the effectome." *Nature* **634**(8032): 201-209.

Schmidt, H., A. Gour, J. Straehle, K. M. Boergens, M. Brecht and M. Helmstaedter (2017). "Axonal synapse sorting in medial entorhinal cortex." *Nature* **549**(7673): 469-475.

Schmidt, M., A. Motta, M. Sievers and M. Helmstaedter (2024). "RoboEM: automated 3D flight tracing for synaptic-resolution connectomics." *Nature Methods* **21**(5): 908-913.

Schneider-Mizell, C. M., A. L. Bodor, D. Brittain, J. Buchanan, D. J. Bumbarger, L. Elabbady, C. Gamlin, D. Kapner, S. Kinn, G. Mahalingam, S. Seshamani, S. Suckow, M. Takeno, R. Torres, W. Yin, S. Dorkenwald, J. A. Bae, M. A. Castro, A. Halageri, Z. Jia, C. Jordan, N. Kemnitz, K. Lee, K. Li, R. Lu, T. Macrina, E. Mitchell, S. S. Mondal, S. Mu, B. Nehoran, S. Popovych, W. Silversmith, N. L. Turner, W. Wong, J. Wu, J. Reimer, A. S. Tolias, H. S. Seung, R. C. Reid, F. Collman and N. M. Da Costa (2025). "Inhibitory specificity from a connectomic census of mouse visual cortex." *Nature* **640**(8058): 448-458.

- Shahmoradian, S. H., E. H. R. Tsai, A. Diaz, M. Guizar-Sicairos, J. Raabe, L. Spycher, M. Britschgi, A. Ruf, H. Stahlberg and M. Holler (2017). "Three-Dimensional Imaging of Biological Tissue by Cryo X-Ray Ptychography." *Sci Rep* **7**(1): 6291.
- Shapson-Coe, A., M. Januszewski, D. R. Berger, A. Pope, Y. Wu, T. Blakely, R. L. Schalek, P. H. Li, S. Wang, J. Maitin-Shepard, N. Karlupia, S. Dorkenwald, E. Sjostedt, L. Leavitt, D. Lee, J. Troidl, F. Collman, L. Bailey, A. Fitzmaurice, R. Kar, B. Field, H. Wu, J. Wagner-Carena, D. Aley, J. Lau, Z. Lin, D. Wei, H. Pfister, A. Peleg, V. Jain and J. W. Lichtman (2024). "A petavoxel fragment of human cerebral cortex reconstructed at nanoscale resolution." *Science* **384**(6696): eadk4858.
- Tran, H. T., E. H. R. Tsai, A. J. Lewis, T. Moors, J. G. J. M. Bol, I. Rostami, A. Diaz, A. J. Jonker, M. Guizar-Sicairos, J. Raabe, H. Stahlberg, W. D. J. Van De Berg, M. Holler and S. H. Shahmoradian (2020). "Alterations in Sub-Axonal Architecture Between Normal Aging and Parkinson's Diseased Human Brains Using Label-Free Cryogenic X-ray Nanotomography." *Frontiers in Neuroscience* **14**.
- Van Harrevel, A. and F. I. Khattab (1967). "Changes in cortical extracellular space during spreading depression investigated with the electron microscope." *J Neurophysiol* **30**(4): 911-929.
- Witvliet, D., B. Mulcahy, J. K. Mitchell, Y. Meirovitch, D. R. Berger, Y. Wu, Y. Liu, W. X. Koh, R. Parvathala, D. Holmyard, R. L. Schalek, N. Shavit, A. D. Chisholm, J. W. Lichtman, A. D. T. Samuel and M. Zhen (2021). "Connectomes across development reveal principles of brain maturation." *Nature* **596**(7871): 257-261.
- Zhang, Y., C. Bosch, T. Ackels, A. Laugros, A. Bonnin, J. Livingstone, C. Waltenberg, M. Berning, S. Tootoonian, M. Kollo, A. Nathansen, N. Rzepka, P. Cloetens, A. Pacureanu and A. Schaefer (2025). "Structure-Function Mapping of Olfactory Bulb Circuits with Synchrotron X-ray Nanotomography." bioRxiv.
- Zhang, W., J. L. Dresselhaus, H. Fleckenstein, M. Prasciolu, M. Zakharova, N. Ivanov, C. Li, O. Yefanov, T. Li, D. Egorov, I. De Gennaro Aquino, P. Middendorf, J. Hagemann, S. Shi, S. Bajt and H. N. Chapman (2024). "Fast and efficient hard X-ray projection imaging below 10 nm resolution." *Opt Express* **32**(17): 30879-30897.